# A GRAPH-BASED SYNTHETIC DATA PIPELINE FOR SCALING HIGH-QUALITY DATA

## ABSTRACT

Synthesizing high-quality data for continual training has been proven to be effective in enhancing the performance of Large Language Models (LLMs). However, previous synthetic approaches struggle to easily scale up data and incur high costs in the pursuit of high quality. In this paper, we propose the Graph-based Synthetic Data Pipeline (GSDP), an economical and scalable framework for high-quality reasoning data synthesis. Inspired by knowledge graphs, we extracted knowledge points from seed data and constructed a knowledge point relationships graph to explore their interconnections. By exploring the implicit relationships among knowledge, our method achieves $\times 255$ data expansion. Furthermore, GSDP led by open-source models, achieves synthesis quality comparable to GPT-4-0613 while maintaining $\times 100$ lower costs. To tackle the most challenging mathematical reasoning task, we present the GSDP-MATH dataset comprising over 1.91 million pairs of math problems and answers. After fine-tuning on GSDP-MATH, *GSDP*-7B based on Mistral-7B achieves 37.7% accuracy on MATH and 78.4% on GSM8K, demonstrating the effectiveness of our method. The dataset and models trained in this paper will be available.

## 1 INTRODUCTION

Despite the remarkable capabilities large language models (LLMs) have demonstrated in various linguistic tasks, significant gaps remain in their ability to comprehend and solve intricate reasoning tasks (e.g., mathematics, coding, physics, and chemistry). One effective approach to bridging these gaps is using large-scale, high-quality synthetic data. However, it is still a challenge to develop a low-cost and effective synthesis pipeline.

Take mathematics as an example. The two main approaches for building high-quality mathematics reasoning datasets are data filtering and data synthesis. Data filtering (Yue et al., 2024b; Shao et al., 2024; Ying et al., 2024) involves extracting data from pre-training corpora such as Common Crawl, and rewriting it using advanced commercial models or human annotation. However, the vast scale and inherent noise of these corpora result in high post-processing costs and inconsistent data quality. Data synthesis (Yu et al., 2023; Luo et al., 2023; Yue et al., 2024a; Tang et al., 2024; Huang et al., 2024a; Li et al., 2024a; Toshniwal et al., 2024; Li et al., 2024b) leverages frontier large language models, such as GPT-3.5 (Floridi & Chiriatti, 2020) and GPT-4 (Achiam et al., 2023), to augment or regenerate high-quality mathematical reasoning datasets. One approach entails rewriting or re-generating similar problems based on seed data for data augmentation. Another approach entails generating new problems using knowledge points. The "knowledge points" refers to fine-grained math concepts (e.g., the Pythagorean theorem, polynomial factorization skills) in problem solving, and they can be generated freshly via LLMs or extracted from existing seed data. Although data synthesis is straightforward, it still suffers from three significant drawbacks: (1) **Limited scalability**: Existing methods have poor scalability, making it difficult to synthesize larger-scale data from smaller seed data. (2) **High cost**: Data synthesis relies on the assistance of closed-source models, which introduces considerable synthesis costs. (3) **Similarity to seed data**: Due to the over-reliance on seed data during synthesis, the new data becomes highly similar to the seed.

To address these drawbacks, we introduce **Graph-based Synthetic Data Pipeline (GSDP)**, a novel framework to scale the synthesis of high-quality data. As presented in Figure 2, GSDP encompasses four critical stages: (1) The knowledge base consists of Knowledge Points (KPs). Its construction

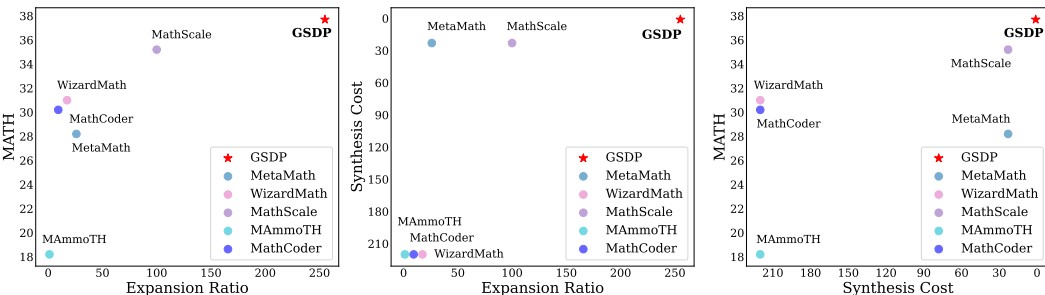

Figure 1: A comparison of various methods in terms of expansion ratio (times), synthesis cost (0.01 cents), and mathematical capability (%). The expansion ratio represents the ratio of the total synthesized data to the total seed data, while the synthesis cost refers to the expenses associated with closed-source models or GPU usage for synthesizing a single data point. The mathematical metrics shown in the figure result from models fine-tuned based on Mistral-7B. As illustrated in the figure, our method demonstrates superior performance across all three dimensions.

starts by extracting KPs from seed data using a specialized mathematical model, followed by a filtering algorithm to remove duplicates and low-quality KPs. (2) We construct a **Knowledge Point Relationships Graph (KPRG)**, where the nodes represent KPs, and the edges indicate that two KPs co-occur in the same problem. We then define relationships between two KPs as follows: pairs of knowledge points one edge apart are explicit, while those more than one edge apart are implicit. (3) Using designed prompts and combinations of KPs selected from the KPRG as input, the mathematical model can generate new questions and solutions. (4) Multiple advanced open-source models are employed to jointly score the synthesized questions and solutions. Ultimately, based on 7500 questions and solutions from the MATH training set as seed data, we synthesized a new dataset comprising over 1.91 million pairs of math questions and solutions, named **GSDP-MATH**.

The highlight of the KPRG is its ability to explore both explicit and implicit relationships among knowledge points. Unlike previous methods that either relied entirely on seed data or focused solely on explicit relationships, our approach leverages both types of connections by KPRG. This allows us to address the drawbacks mentioned above: (1) **High scalability**: By utilizing implicit relationships, we can synthesize a much larger volume of data, as implicit relationships are far more abundant than explicit ones. (2) **Low seed similarity**: Since implicit relationships do not appear in the seed data, they help synthesize problems that are different from the seed data. (3) **Low cost**: The entire pipeline uses only open-source models, significantly reducing synthesis costs.

By comparing the expansion ratio, synthesis cost, and mathematical capability in Figure 1, and the seed similarity and data diversity in Appendix E, we demonstrate the advantages of our method from a quantitative perspective. Specifically, GSDP achieves an expansion ratio of up to 255, incurs less than 1% of the cost compared to other methods, and GSDP-MATH exhibits lower seed similarity and greater diversity. Moreover, we validate the effectiveness of GSDP-MATH on several base models, including Mistral-7B (Jiang et al., 2023), LLaMA3-8B (Meta, 2024), and Qwen1.5-7B (Bai et al., 2023). The GSDP models demonstrably surpass the base models across four mathematical reasoning benchmarks: MATH (Hendrycks et al., 2021), GSM8K (Cobbe et al., 2021), Gaokao-Bench (Team, 2024), and SVAMP (Patel et al., 2021). Notably, *GSDP*-7B based on Mistral-7B achieves 78.4% on GSM8K and 37.7% on MATH, surpassing all competitors under the same conditions.

## 2 PROPOSED METHOD

### 2.1 OVERVIEW

This section introduces Graph-based Synthetic Data Pipeline (GSDP) is unified synthetic data framework with four parts containing: Knowledge Base Construction, Knowledge Point Relationships Graph Construction, Synthesis Based on Diverse Knowledge Point Combinations, and Evaluation of Problems and Solutions, as illustrated in Figure 2. Detailed descriptions and implementation steps for each component are provided in the subsequent sections.

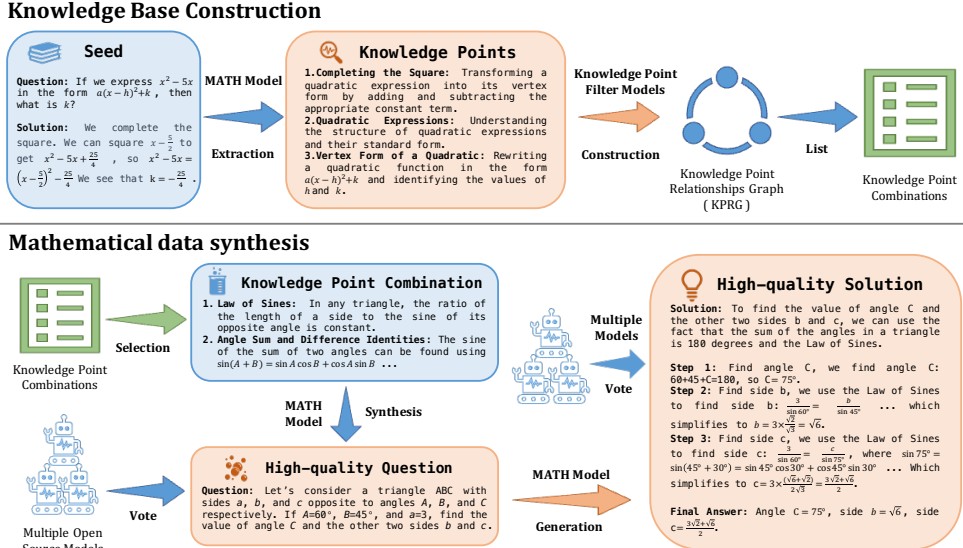

Figure 2: The overview of the Graph-based Synthetic Data Pipeline (GSDP). GSDP begins with seed math data and follows a four-step process: (1) knowledge base construction, (2) knowledge point relationship graph construction, (3) graph-based synthesis, and (4) evaluation by multiple models voting. After these steps, we obtain the GSDP-MATH dataset, which is subsequently used to train open LLMs. Finally, we obtain GSDP models.

## 2.2 KNOWLEDGE BASE CONSTRUCTION

In order to help LLMs better understand complex information like math problems, we decompose seed data into simpler meta information and then construct a knowledge base to represent the entire dataset in a more flexible form. The meta information contained in a problem includes elements such as "Subject", "Topic", and "Knowledge Point". For instance, the "Subject" might be "Mathematics", the "Topic" could be "Algebra", and the "Knowledge Point" could be "methods for solving quadratic equations". To simplify the extraction process and facilitate comprehension by the model, the meta information we extract is limited to "Knowledge Point", and we demonstrate through experiments that "Knowledge Points" sufficiently encapsulate most of the information about a problem.

GSDP use the MATH training set as the seed data which consist of 7.5k math problems. As shown in Figure 2, we first extract no more than 10 relevant knowledge points (KPs) from each seed problem with prompt engineering of DeepSeek-Math-RL (Shao et al., 2024) (see Prompt 2.1 for the prompt). After extracting the KPs, we employ an embedding model (Chen et al., 2023a) and a large language model (LLM) for dual filtering of the KPs. Initially, the LLM filters out KPs with vagueness, mathematical errors, or excessive detail. Subsequently, the embedding model clusters KPs with similar meanings, followed by a second validation using the LLM. Finally, the most appropriate and accurate KP from each cluster is chosen to represent the cluster. For more details on dual filtering and KP examples, please refer to Appendix D.

---

**Prompt 2.1 : Prompt for Knowledge Points Extraction**

As a mathematics education specialist, please analyze the given math problem and its solution to **extract specific mathematical knowledge points**. ...

Please follow these requirements: (1) **Extract Knowledge Points**: Identify relevant mathematical knowledge points from the given problem and its solution. (2) **Ensure Relevance**: Make sure the extracted knowledge points are directly related to the problem, precise, and concise. Avoid vague concepts. (3) **Focus on Key Concepts**: Concentrate on the specific concepts essential for solving the problem and explaining the solution. (4) **Provide a Clear and Concise List**: : Offer a clear, succinct list of the knowledge points so that educators can design related exercises, helping students focus on the critical learning outcomes needed for mastering the subject.

---

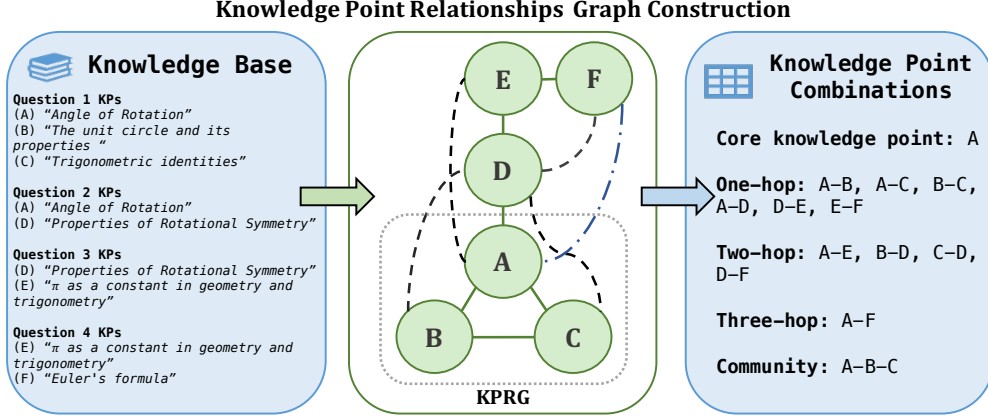

Figure 3: An example of constructing the Knowledge Point Relationships Graph (KPRG) from an existing knowledge base and identifying the four knowledge points combination we proposed.

## 2.3 KPRG CONSTRUCTION

To structure the disordered KPs in the knowledge base and explore their specific interconnections, we designed a Knowledge Point Relationships Graph (KPRG) to get all related KPs pairs.

In the KPRG, each node is represented as a KP, and each edge represents that the connected KPs have co-occurred in the same problem. Specifically, the KPRG $\mathcal{G}$ can be represented as $\mathcal{G} = (\mathbb{K}, \mathbb{E})$. The nodes $\mathbb{K}$, which refer to knowledge points, are denoted as $\mathbb{K} = \{\mathbf{k}_1, \mathbf{k}_2, \ldots, \mathbf{k}_{|\mathbb{K}|}\}$. The edges $\mathbb{E}$ are denoted as $\mathbb{E} = \{\mathbb{E}_{\text{ex}}, \mathbb{E}_{\text{im}}\}$, and there are two types: (1) Explicit ($\mathbb{E}_{\text{ex}}$): Knowledge points that appear together in the same seed problem are connected by a solid edge. The explicit edges $\mathbb{E}_{\text{ex}}$ can be denoted as $\mathbb{E}_{\text{ex}} = \{(\mathbf{k}_i, \mathbf{k}_j) | \mathbf{D}(\mathbf{k}_i, \mathbf{k}_j) = 1\}$, where the edge distance $\mathbf{D}(\mathbf{k}_i, \mathbf{k}_j)$ represents the number of solid edges in the shortest path between $\mathbf{k}_i$ and $\mathbf{k}_j$. Additionally, the edge weight $\mathbf{W}(\mathbf{k}_i, \mathbf{k}_j)$ is recorded to denote the co-occurrence frequency between $\mathbf{k}_i$ and $\mathbf{k}_j$. (2) Implicit ($\mathbb{E}_{\text{im}}$): Knowledge points with more than one solid edge between them are connected by a dashed edge, as shown in Figure 3. The implicit edges $\mathbb{E}_{\text{im}}$ can be denoted as $\mathbb{E}_{\text{im}} = \{(\mathbf{k}_i, \mathbf{k}_j) | \mathbf{D}(\mathbf{k}_i, \mathbf{k}_j) > 1\}$.

## 2.4 SYNTHESIS BASED ON DIVERSE KNOWLEDGE POINT COMBINATIONS

Different from previous methods that primarily explored explicit relationships between KPs, we proposed four types of knowledge point relationships: one-hop, two-hop, three-hop, and community, which fully integrate both explicit and implicit relationships to generate more diverse synthetic data. Additionally, the exploration of implicit relationships has helped us obtain more combinations of knowledge points, which is also a key reason why GSDP can achieve a high expansion ratio. Figure 3 illustrates an example of constructing KPRG from an existing knowledge base and identifying these four KPs combinations.

**One-hop** represents explicit relationships, consisting of all pairs of KPs that are directly connected by a single edge. The weight of the edge represents the frequency of their co-occurrence. One-hop combinations have appeared in the seed data, thus they have high relevance.

**Two-hop** is a type of implicit relationship, consisting of all pairs of KPs that are indirectly connected by two edges. Two-hop is the nearest implicit relationship, connected indirectly through one node, maintaining a certain degree of relevance.

**Three-hop** further explores implicit relationship, consisting of all pairs of KPs indirectly connected through three edges, with one core knowledge point and one other knowledge point in each pair. Core KPs are those with the highest number of connections in the KPRG, indicating their importance and wide applicability within the knowledge system. When the graph is large, there may be multiple core KPs. As the distance between knowledge points increases, the relevance tends to weaken, so we focus on core knowledge points to ensure that Three-hop remains meaningful. For example, assume

that a core knowledge point is "calculus". In the seed data, problems only associate "calculus" with fundamental concepts such as "limits" and "derivatives". However, in the KPRG, "calculus" is likely not adjacent but close to KPs such as "Fourier transforms" and "complex functions". By integrating these three-hop KPs to construct new problems (the same applies to two-hop), it can significantly increase the diversity of the problems and simultaneously promote the model's ability to learn new types of problems, thereby enhancing its mathematical capabilities.

**Community** represents explicit relationships, consisting of three knowledge points, each pair of which is mutually connected by edges. There is a strong correlation between the three KPs.

Accordingly, the one-hop combinations are used to synthesize high-quality variant problems directly related to the seed data. Implicit relationships combinations are used to synthesize new distribution data, increasing the diversity of the dataset. Community combinations are utilized to generate complex and challenging problems, enriching the dataset's difficulty levels.

After listing all valid knowledge point combinations, we input the prompt and knowledge point combinations into the math model to synthesize new problems. We do not include few-shot examples in the prompt, as this would cause the model to generate problems too similar to them. The prompt is shown in Prompt 2.2. Before solution generation, a rating model assigns a difficulty level to each problem. For medium and low difficulty issues, DeepSeek-Math-RL generates the solutions, while LLaMA3.1-70B (Dubey et al., 2024) handles high difficulty problems.

Moreover, we implement a decontamination process to remove all math problems found in the MATH dataset.

---

**Prompt 2.2 : Prompt for New Problem Generation**

You are a math teacher. Now, you need to help your students learn the following math knowledge points. Using these knowledge points as guidelines, **please construct a new, original math problem** that requires an understanding and application of all these points. Ensure the following: (1) The constructed problem must be **free from any mathematical logic errors**. (2) The problem must **combine all the knowledge points**. (3) The question should be of **sufficient difficulty** and **logically consistent**.

---

### 2.5 EVALUATION OF PROBLEMS AND SOLUTIONS

To achieve the same effectiveness as closed-source models, we use multiple open-source models jointly scoring to filter the data. The more models involved, the higher the data quality, but this also increases the amount of data filtered out and the time required. Therefore, we need to find an optimal combination of open-source models to balance high-quality rate and total data volume. In Experiment 3.8, supervised by GPT-4, we find a combination that achieves 94% of GPT-4-0613's effectiveness while retaining 45% of the data: Qwen2 (Yang et al., 2024), InternLM2 (Cai et al., 2024), and LLaMA3.1 (Dubey et al., 2024).

For problems evaluation, we use a weighted scoring filtering strategy. Problems are evaluated on two criteria: logical completeness (absence of mathematical errors and accurate relation to provided knowledge points) and presentational completeness (clarity, completeness, and absence of prompts or answers). We use the multiple open-source models for evaluation, assigning each problem a score between 0 and 1 based on a weighted sum of model scores (with a threshold of 0.85). For solutions evaluation, we adopt a single-vote veto strategy. The evaluation model requires solutions to be mathematically error-free and to fully satisfy the problem requirements. Each solution is scored as either 0 or 1, depending on whether all models unanimously agree on its correctness. Solutions flagged as problematic by any model are filtered out. Finally, we constructed a dataset named GSDP-MATH comprising 1.91 million high-quality mathematical questions at low cost.

## 3 EXPERIMENTS

### 3.1 EXPERIMENTAL SETUP

Due to the exceptional scalability of GSDP, we are able to synthesize a substantial quantity of high-quality data with minimal seed data, rendering the GSDP-MATH dataset suited for pre-training and

Table 1: The performance of models on mathematical reasoning tasks. Results are sourced from MAmmoTH2 and OpenCompass. GAOKAO II denotes the 2010-2022 Math II MCQs from GAOKAO-Eval, and GAOKAO I represents the 2010-2022 Math I MCQs. MathScale Mistral (Official) indicates that the official model has not been open-sourced but has provided results on the MATH and GSM8K datasets. MathScale Mistral (Reproduced) refers to MathScale Mistral reproduced by others. The **bold** and underlined denote the first and second, respectively.

| Model | MATH | GSM8K | GAOKAO II | GAOKAO I | SVAMP | AVG |
|---|---|---|---|---|---|---|
| *Closed-source Models* | | | | | | |
| GPT-4-0613 | 42.5 | 92.0 | - | - | 93.1 | - |
| GPT-3.5-Turbo | 37.8 | 74.1 | - | - | - | - |
| *Open-Source Models with Parameter Sizes of 7B or 8B* | | | | | | |
| Yi-6B | 6.6 | 39.6 | 6.3 | 3.1 | 55.5 | 22.2 |
| LLaMA2-7B | 3.6 | 16.8 | 16.9 | 16.4 | 38.0 | 18.3 |
| Qwen1.5-7B | 13.3 | 54.1 | 56.4 | 53.7 | 73.4 | 50.2 |
| LLaMA3-8B | 21.3 | 54.8 | 4.1 | 7.9 | 69.7 | 31.6 |
| Mistral-7B | 11.2 | 36.2 | 13.8 | 12.2 | 66.9 | 28.0 |
| InternLM2-7B | 25.2 | 69.9 | 33.9 | 35.5 | 71.5 | 47.2 |
| Deepseek-7B | 6.4 | 17.4 | 14.2 | 13.1 | 46.2 | 19.5 |
| Deepseek-Math-7B | 36.2 | 64.2 | 34.9 | 28.9 | 73.2 | 47.5 |
| MetaMath Mistral | 28.2 | 77.7 | 9.2 | 9.4 | 77.2 | 40.3 |
| WizardMath v1.1 | 31.0 | 78.0 | 17.0 | 15.4 | 48.5 | 38.0 |
| MathCoder-CL-7B | 30.2 | 67.8 | 9.6 | 15.9 | 70.7 | 38.8 |
| MathScale Mistral (Official) | 35.2 | 74.8 | - | - | - | - |
| MathScale Mistral (Reproduced) | 34.5 | 74.0 | 36.7 | 31.3 | 79.6 | 51.2 |
| MAmmoTH-Mistral-7B | 18.2 | 61.5 | 22.0 | 21.5 | 71.7 | 39.0 |
| MAmmoTH2-7B | 36.7 | 68.4 | 44.9 | 29.4 | 81.8 | 52.2 |
| MAmmoTH2-8B | 35.8 | 70.4 | 33.5 | 24.3 | 78.6 | 48.5 |
| *Trained only with GSDP-MATH* | | | | | | |
| **GSDP-Qwen-7B** | 36.8 | 73.4 | **56.8** | **55.1** | 79.9 | **60.4** |
| Δ over Qwen1.5 | +23.5 | +19.3 | +0.4 | +1.4 | +6.5 | +10.2 |
| **GSDP-8B** | 37.2 | 76.5 | 38.5 | 31.8 | 82.2 | 53.2 |
| Δ over LLaMA3 | +15.9 | +21.7 | +34.4 | +23.9 | +12.5 | +21.6 |
| **GSDP-7B** | **37.7** | **78.4** | 40.8 | 31.3 | **82.3** | 54.1 |
| Δ over Mistral | +26.5 | +42.2 | +27.0 | +19.1 | +15.4 | +26 |

instruction fine-tuning tasks. For our experiments, we selected the open-source models Qwen1.5-7B (Bai et al., 2023), Mistral-7B (Jiang et al., 2023), and LLaMA3-8B (Meta, 2024)as the baseline models. We call the fine-tuned models *GSDP-Qwen*-7B, *GSDP*-8B, and *GSDP*-7B, based on Qwen1.5-7B, LLaMA3-8B, and Mistral-7B, respectively.

In the instruction fine-tuning experiment, we only employ GSDP-MATH data and fine-tune the models using the LLaMAFactory (Zheng et al., 2024) framework. The fine-tuning task was conducted for 2 epochs with a learning rate of 5e-6, a global batch size of 128, and a maximum sequence length of 2048. A cosine schedule with a 3% warm-up ratio is adopted to regulate the learning rate. For expedited and efficient training, we leveraged DeepSpeed (Rasley et al., 2020) ZeRO Stage 3 and FlashAttention 2 (Dao, 2023).

In the pre-training experiment, we first reformatted the mathematical problems and solutions from GSDP-MATH into the following template: "Problem:\n{probelm}\n\nSolution:\n{solution}". Then we combined GSDP-MATH with publicly available pre-training data for this experiment. To validate the efficacy of our dataset, we utilized the Megatron-lm (Shoeybi et al., 2019) framework for model pre-training. The training configuration included a learning rate of 2e-5, a global batch size of 512, a maximum sequence length of 4096, and a total of 3.5 billion tokens trained.

Table 2: Comparison of various methods in expansion ratio and synthesis cost. Synthesis Cost indicates the expenses associated with closed-source models or GPU usage for synthesizing a single data point. The expansion ratio represents the proportion of the total synthesized data to the total seed data. The notation (1B) indicates that the dataset contains approximately 1 billion tokens. TS refers to Total Seed Data. TSD refers to Total Synthesized Data. ER represents the Expansion Ratio (times). SC represents the Synthesis Cost (0.01 cents).

| Method | Data Source | Synthesis Model | TS | TSD | ER | SC |
|---|---|---|---|---|---|---|
| MetaMath (Yu et al., 2023) | GSM8K+MATH | GPT-3.5 | 15K | 395K | 26 | 23 |
| MathScale (Tang et al., 2024) | MWPBENCH | GPT-3.5 | 20K | 2M | 100 | 23 |
| WizardMath (Luo et al., 2023) | GSM8K+MATH | GPT-4 | 15K | 96K | 6.4 | 220 |
| XwinMath (Li et al., 2024a) | GSM8K+MATH | GPT-4 | 15k | 1.4M | 93 | 220 |
| MAmmoTH (Yue et al., 2024a) | MAmmoTH datasets | GPT-4 | 220K | 262K | 1.2 | 220 |
| MathCoder (Wang et al., 2023) | GSM8K+MATH | GPT-4 | 15K | 80K | 5.3 | 220 |
| **GSDP** | MATH | Open-Source Model | 7.5K | **1.9**M(1B) | **255** | **1.23** |

## 3.2 EVALUATION DATASETS

To rigorously assess the enhancement in mathematical reasoning capabilities of models trained with GSDP-MATH, we employed a suite of mathematical evaluation datasets, including GSM8K (Cobbe et al., 2021), MATH (Hendrycks et al., 2021), GAOKAO-Eval (Team, 2023) and SVAMP (Patel et al., 2021). Additionally, to evaluate the pre-trained models' performance across pre-training tasks, we utilized evaluation datasets such as MMLU (Hendrycks et al., 2020), C-Eval (Huang et al., 2024c), and MBPP (Austin et al., 2021). We employed testing scripts provided by MAmmoTH2 (Yue et al., 2024b) and OpenCompass (Contributors, 2023).

## 3.3 MAIN RESULTS

Table 1 presents the main results of our fine-tuned models on mathematical reasoning tasks. It is evident that models trained only on the GSDP-MATH dataset exhibit substantial improvements over other models based on Mistral-7B and LLaMA3-8B. For example, the performance of *GSDP*-7B achieves 37.7% accuracy on MATH and 78.4% on GSM8K, surpassing other competitive counterparts, including MAmmoTH2-7B (Yue et al., 2024b), MathScale Mistral (Reproduced)[1], and others. Meanwhile, GSDP-8B and GSDP-Qwen-7B boost the performance of LLaMA3-8B and Qwen1.5-7B by an average of 22 and 10 points, respectively. This demonstrates that GSDP does not rely on the base model and has advanced generalization capabilities. Notably, even for the GSM8K, GAOKAO, and SVAMP datasets that were not in the seed data or training set, our model still significantly outperforms other models on these out-of-domain test sets.

## 3.4 COMPARISON WITH OTHER METHODS

To elucidate the advantages of our methodology, we compare various mathematical data construction methods in Table 2. It can be observed that all approaches necessitate the utilization of GPT-3.5 or GPT-4. Although employing closed-source models can ensure the quality of synthesized data to some extent, it inevitably incurs substantial costs and limits the total quantity of synthesized data. For expansion ratio, MathScale (Tang et al., 2024) represents the largest dataset, leveraging GPT-3.5 to synthesize 2M new data points from a 20K seed dataset, achieving an expansion ratio of 100-fold. However, methods (Luo et al., 2023; Yue et al., 2024a) utilizing GPT-4 for data synthesis exhibit relatively low expansion ratios due to the high costs involved. In contrast, GSDP demonstrates exceptional scalability, achieving a 255-fold increase. Based on a mere 7.5K seed dataset, it synthesizes 1.9M high-quality data points containing 1B tokens. Due to its outstanding scalability, GSDP can be used to generate pre-training datasets. Compared to publicly available pre-training datasets, data generated from GSDP contains less noise, higher quality, and greater controllability. For synthesis cost, we posit that methods using closed-source models incur cost solely from the

---

[1]https://huggingface.co/fdqerq22ds/MathScale-Mistral

Table 3: Main results on pre-training tasks. The **bold** and underlined denote the first and second, respectively. The results in the table are sourced from OpenCompass.

| Model | MATH | GSM8K | MMLU | CEVAL | MBPP | AVG |
|---|---|---|---|---|---|---|
| Yi-6B | 6.6 | 39.6 | 64.1 | 70.78 | 30.4 | 42.3 |
| Qwen-7B | 15.6 | 54.4 | 59.8 | 62.06 | 46.7 | 47.7 |
| LLaMA2-7B | 3.6 | 16.8 | 46.8 | 30.13 | 26.5 | 24.8 |
| Deepseek-7B | 6.4 | 17.4 | 49.2 | 43.68 | 44.0 | 32.1 |
| InternLM2-7B | 25.2 | 69.9 | 65.8 | 63.5 | 54.47 | 55.8 |
| Mistral-7B | 11.2 | 36.2 | 64.0 | 43.76 | 47.47 | 40.5 |
| Qwen1.5-7B | 13.3 | 54.1 | 62.2 | **72.49** | 52.14 | 50.8 |
| *Pre-training on LLaMA3-8B* | | | | | | |
| LLaMA3-8B | 21.3 | 54.8 | **66.4** | 48.8 | 54.9 | 49.2 |
| **GSDP-LLaMA3-8B** | **37.8** | **73.5** | 66.2 | 51.2 | **56.8** | **57.1** |
| Δ over LLaMA3 | +16.5 | +18.7 | -0.2 | +2.4 | +1.9 | +7.9 |

closed-source model cost[2]; whereas for our method, we only need account for GPU usage cost[3]. More details on the calculation of synthesis costs can be found in Appendix B. As shown in Table 2, the cost of synthesizing a single data point using GSDP is 5% or even less than 1% of the cost incurred by other methods. For data quality, we compared the seed similarity and data diversity of various methods by embedding the datasets. The results in Appendix E show that GSDP-MATH exhibits lower seed similarity and greater diversity. Additionally, our model demonstrated highly competitive mathematical capabilities after being fine-tuned on the GSDP-MATH dataset.

## 3.5 RESULTS OF PRE-TRAINED MODELS

Table 3 outlines the results of our pre-trained model across multiple pre-training tasks. Owing to the excellent scalability of GSDP, the GSDP-MATH dataset is fully adequate for pre-training. We integrated the GSDP-MATH dataset with other publicly available pre-training data to pre-train the LLaMA3-8B base model, which we call *GSDP-LLaMA3*-8B. During this experiment, the model demonstrated significant improvements in mathematical capabilities while maintaining stability in other general capabilities. The *GSDP-LLaMA3*-8B exhibited improvements of 16.5% and 18.7% on the MATH and GSM8K benchmarks, respectively. Moreover, it maintained stable performance on MMLU, CEVAL and MBPP. On average, our model achieved a 7.9% improvement, outperforming most open-source models with similar parameter sizes.

## 3.6 GENERAL SCIENTIFIC REASONING ABILITY EVALUATION

GSDP-MATH not only enhances the model's mathematical reasoning capabilities but also has a significant positive impact on its performance in out-of-domain reasoning tasks. In addition to mathematical reasoning, we use several widely-used datasets to test the model's scientific reasoning ability in subjects such as physics, biology, chemistry, and computer science. These datasets include ARC-C (Clark et al., 2018), MMLU-STEM (Hendrycks et al., 2021), GPQA (Rein et al., 2023), BBH (Suzgun et al., 2022), TheoremQA (Chen et al., 2023b), and MBPP (Austin et al., 2021). The experimental results in Appendix C show that our three GSDP models achieve an average improvement of over 5% compared to the base models across all scientific reasoning tasks. Moreover, they perform exceptionally well on multiple benchmarks when compared to other mathematical models.

## 3.7 COMPARATIVE ANALYSIS OF DIVERSIFIED KNOWLEDGE POINTS COMBINATION

To clarify the impact of explicit and implict knowledge point combinations on model performance, we partitioned the GSDP-MATH dataset into four distinct segments: (1) GSDP-One: This seg-

---

[2]https://openai.com/api/pricing/
[3]https://power.netmind.ai/rentIntro

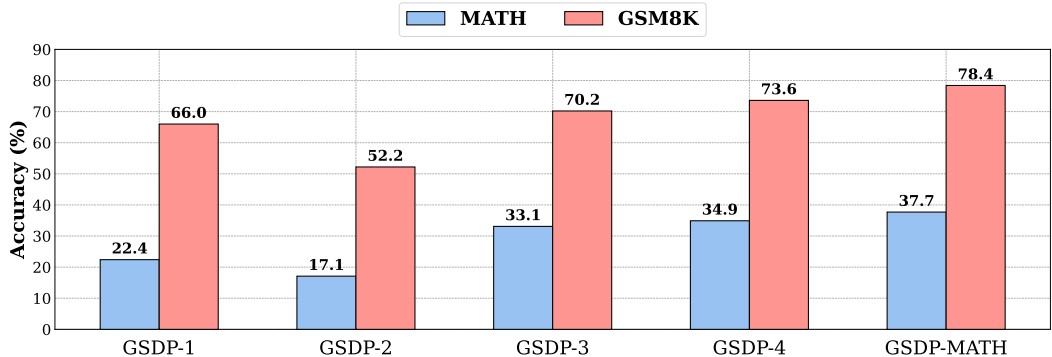

Figure 4: Performance comparison of different data components. The results for MATH and GSM8K are derived from models fine-tuned on Mistral-7B. GSDP-1 denotes the model based on Mistral-7B and fine-tuned with GSDP-One; GSDP-2 denotes GSDP-One-Base; GSDP-3 denotes GSDP-Two + GSDP-Three; GSDP-4 denotes GSDP-One + GSDP-Two + GSDP-Three; GSDP-MATH denotes GSDP-One + GSDP-Two + GSDP-Three + GSDP-Community.

ment consists of data generated via combinations of one-hop knowledge points, with additional data generated based on edge weight repetition. GSDP-One-Base represents data where each one-hop knowledge point combination is generated only once. (2) GSDP-Two: This segment encompasses data derived from combinations of two-hop knowledge points. (3) GSDP-Three: This segment includes data generated from combinations of three-hop knowledge points. (4) GSDP-Community: This segment features data generated from combinations of community knowledge points.

We used different data combinations for training Mistral-7B, and the results are shown in Figure 4. Using GSDP-One enhances model performance more than using GSDP-One-Base because GSDP-One includes more variant questions. Adding GSDP-Two and GSDP-Three significantly boosts model performance because these datasets enable the model to learn from a broader distribution of questions. We also found that incorporating GSDP-Community further enhances model performance, as this dataset mainly contains more challenging questions, thereby improving the model's ability to solve complex mathematical problems.

## 3.8 JOINT SCORING EXPERIMENT

Table 4: Accuracy, precision, recall, F1-score and retention ratio of predictions made by various combinations of open-source models, assuming GPT-4's predictions as the ground truth. Retention ratio refers to the proportion of data retained after filtering relative to the total data.

| Exp.ID | InternLM2-20B | Qwen2-14B | LLaMA3.1-8B | Yi-34B | Accuracy | Precision | Recall | F1-score | Retention Ratio |
|---|---|---|---|---|---|---|---|---|---|
| 1 | ✔ | | | | 0.29 | 0.35 | 0.50 | 0.41 | 0.71 |
| 2 | ✔ | ✔ | | | 0.80 | 0.75 | 0.89 | 0.81 | 0.65 |
| 3 | ✔ | ✔ | | ✔ | 0.87 | 0.85 | **0.90** | 0.87 | 0.59 |
| 4 | ✔ | ✔ | ✔ | | **0.90** | 0.94 | 0.85 | **0.89** | 0.45 |
| 5 | ✔ | ✔ | ✔ | ✔ | 0.75 | **0.97** | 0.52 | 0.68 | 0.27 |

To compare the evaluation performance of the joint scoring model with GPT-4-0613, we constructed a test set comprising 5000 mathematical problems with solutions. The dataset is selected from the synthetic data before evaluation. Each data point is annotated with two labels by GPT-4 using Prompt A.5 and Prompt A.6: problem correctness and solution correctness. Subsequently, we apply the joint scoring model to evaluate these problems and their solutions with the same prompts. We consider the results scored by GPT-4 as the ground truth, while the results scored by the joint scoring model serve as the predicted answers.

We set the candidate pool as Qwen2-14B, InternLM2-20B, LLaMA3.1-8B, and Yi-34B (Young et al., 2024), and then selected several models from this pool to form the joint scoring model. In

evaluating problem correctness, the joint scoring result is a weighted aggregate of individual model scores, with the weights determined based on the mathematical proficiency of each model. For solution correctness evaluation, the joint scoring result takes the lowest score among the models to rigorously ensure the accuracy of the solutions. As shown in Table 4, we comprehensively compared various combinations of accuracy, precision, recall, F1-score and retention ratio. Since we aim to retain as many positive examples as possible in the final data, we should first choose the combination with the highest precision. The combination in Exp.5 demonstrates a precision of 97%, but it performs poorly in recall, which results in retaining only 27% of the data. To balance precision and retention ratio, we chose the more stable combination in Exp.4, which has a precision of 94% while retaining nearly half of the data.

## 4 RELATED WORK

### 4.1 LLMS AND MATHEMATICAL REASONING

Recent research has focused extensively on enhancing the reasoning capabilities of foundational large language models (LLMs). However, general-purpose LLMs such as LLaMA3 (Meta, 2024) , Mistral (Jiang et al., 2023), InternLM2 (Team, 2023), Qwen (Bai et al., 2023) , Yi (Young et al., 2024), and DeepSeek (Bi et al., 2024) have demonstrated suboptimal performance in mathematical reasoning tasks. Consequently, researchers have explored various strategies to augment the mathematical reasoning capabilities of LLMs. The primary approaches encompass continued pre-training and instruction fine-tuning. Continued pre-training involves training LLMs on extensive pre-training datasets (Lewkowycz et al., 2022; Taylor et al., 2022; Azerbayev et al., 2023; Shao et al., 2024; Ying et al., 2024). In contrast, instruction fine-tuning focuses on applying supervised fine-tuning losses to small-scale, high-quality instruction-response pairs (Ouyang et al., 2022; Chung et al., 2024). Both approaches necessitate the availability of high-quality mathematical inference data to be effective.

### 4.2 DATA SYNTHESIS

In the domain of mathematical reasoning, data synthesis is predominantly utilized for instruction fine-tuning, where each data sample comprises a question text and its corresponding answer text. Leveraging our method's exceptional scalability, we are capable of synthesizing extensive quantities of high-quality mathematical inference data from a limited amount of seed data, thus making it equally suitable for continued pre-training tasks. Research efforts primarily concentrate on two pivotal aspects: improving data quality and generating novel questions. Regarding the generation of novel questions, one approach (Yu et al., 2023; Yue et al., 2024a; Tang et al., 2024; Li et al., 2024a; Toshniwal et al., 2024) entails rewriting or generating similar questions based on seed data for data augmentation. Another approach (Huang et al., 2024a;b; Li et al., 2024b) involves generating new questions using knowledge points, either by generating new knowledge points via GPT-4 or extracting them from existing knowledge point databases. However, this approach often suffers from limited scalability, high cost, and high similarity to the seed data, because it primarily focuses on the explicit relationships between KPs and uses GPT-4 or GPT-3.5. Our method overcomes these limitations by exploring both explicit and implicit relationships between KPs with the help of KPRG, and using open-source models instead of closed-source models to synthesize data at low cost.

## 5 CONCLUSION

In this paper, we introduce GSDP, a low-cost, highly scalable paradigm for synthesizing high-quality data. Utilizing this method in mathematics, we constructed the GSDP-MATH dataset, comprising 1.91 million high-quality question-answer pairs specifically designed to enhance the mathematical reasoning capabilities of large language models. By leveraging this dataset, *GSDP*-7B have demonstrated outstanding performance across multiple mathematical test sets. Our research indicates that thoroughly exploring the implicit relationships between knowledge points is an effective method for synthesizing larger-scale and more diverse data. Furthermore, combining multiple open-source models can achieve performance close to closed-source models, which is key to the low-cost synthesis of high-quality data.

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

# A   PROMPTS

---

**Prompt A.1 : Prompt for Knowledge Points Extraction**

As a mathematics education specialist, please analyze the given math problem and its solution to **extract specific mathematical knowledge points**. These knowledge points will assist teachers in creating similar exercises and help students understand and master key learning objectives.

Please follow these requirements:
(1) **Extract Knowledge Points**: Identify relevant mathematical knowledge points from the given problem and its solution.
(2) **Ensure Relevance**: Make sure the extracted knowledge points are directly related to the problem, precise, and concise. Avoid vague concepts.
(3) **Focus on Key Concepts**: Concentrate on the specific concepts essential for solving the problem and explaining the solution.
(4) **Provide a Clear and Concise List**: : Offer a clear, succinct list of the knowledge points so that educators can design related exercises, helping students focus on the critical learning outcomes needed for mastering the subject.

**Math Problem:** {question}
**Solution:** {solution}

Limit the list to no more than ten knowledge points, ensuring knowledge points listed are strictly pertinent to solving the given math problem and aiding in its conceptual comprehension. Please output in this format:

**Relevant Math knowledge points:**

1.

2.

3.

4.  (Continue the list as necessary)

---

**Prompt A.2 : Prompt for Cluster Knowledge Points**

Given a list of similar math knowledge points, pick the one that best represents all of them. This knowledge point should be the most commonly known and used in math discussions, making sure it includes all the listed knowledge points.

**List of knowledge points:** {knowledge points}

**How to Choose:**

1. **Common Usage:** The knowledge points should be widely used in schools and academic settings.

2. **Broad Coverage:** It should cover the key aspects and details of all similar knowledge points listed.

3. **Standard Terminology:** The terms used should be standard and widely accepted in the math community.

Please review each knowledge points and pick the one that fits these criteria best. Provide a short explanation for your choice.
**Format your response like this:**

**Best knowledge points:** {Your choice here}
**Reason:** {Your explanation here}

---

**Prompt A.3 : Prompt for Solution Generation**

{Problem}
Please reason step by step, and put your final answer within \boxed{}.

---

---

**Prompt A.4 : Prompt for New Problem Generation**

You are a math teacher. Now, you need to help your students learn the following math knowledge points. Using these knowledge points as guidelines, **please construct a new, original math problem** that requires an understanding and application of all these points.

Ensure the following:
(1) The constructed problem must be **free from any mathematical logic errors**.
(2) The problem must **combine all the knowledge points**.
(3) The question should be of **sufficient difficulty** and **logically consistent**.

**knowledge points 1:** {knowledge points1}
**knowledge points 2:** {knowledge points2}
**knowledge points 3:** {knowledge points3}

**Please format your response like this:**

**New Problem:** {Your new problem here}

---

**Prompt A.5 : Prompt for Problem evaluation**

Given these math knowledge points:

**knowledge points 1:** knowledge points
**knowledge points 2:** knowledge points

I have formulated a new math problem as follows: {question}

Could you please evaluate whether the new math problem effectively incorporates both of the provided knowledge points and identify any factual or logical errors in the problem? Provide a score as a floating-point number between 0 and 1, where 1 means the problem effectively covers both knowledge points without any factual or logical errors, and 0 indicates that it does not effectively cover any of the knowledge points or there are factual or logical errors.

Please strictly follow the above requirements to give a reasonable score. The format is as follows:

**Evaluation Score:** {score}
**Explanation:** {Your explanation here}

---

**Prompt A.6 : Prompt for Solution evaluation**

You are given a mathematical problem along with its solution. Your task is to determine whether the provided solution is correct. Please read the problem and solution carefully before answering, considering the following criteria:

1. **Accuracy of Calculations:** Check all the numerical calculations to ensure they are carried out correctly.

2. **Logical Consistency:** Verify that the logical steps follow each other coherently and correctly.

3. **Completeness of the Solution:** Confirm that all parts of the problem have been addressed and the solution is comprehensive.

If any of these criteria are not satisfied, you should respond with "False".

**Problem:** {question}
**Solution:** {solution}

Is the provided solution correct? Please follow this format in your response:

**Answer:** {Your judgment here, True or False}
**Explanation:** {Your explanation here}

## B  CALCULATION OF SYNTHESIS COST

For synthesis cost, we posit that methods using closed-source models incur cost solely from the closed-source model cost[4]; whereas for our method, we only need account for GPU usage cost[5]. Based on information from the web pages, the input cost of GPT-4 is \$10 per 1M tokens, and the output cost is \$30 per 1M tokens. For GPT-3.5, the input cost is \$1.5 per 1M tokens and the output cost is \$2 per 1M tokens. The cost of using one NVIDIA RTX 4090 (24G) is \$0.35 per hour.

According to our experiments, synthesizing and scoring problems and solutions requires at least 1000 input tokens and 400 output tokens (with slight differences between various methods). For data synthesis using GPT-4, the cost of synthesizing one data point is calculated as:

$$10 \times 0.001 + 30 \times 0.0004 = 0.022 \, \$$$

In terms of 0.01 cents, the synthesis cost is 220.

For data synthesis using GPT-3.5, the cost of synthesizing one data point is calculated as:

$$1.5 \times 0.001 + 2 \times 0.0004 = 0.0023 \, \$$$

In terms of 0.01 cents, the synthesis cost is 23.

For GSDP, we leveraged the vLLM (Kwon et al., 2023) and used 8 RTX 4090 GPUs for 84 hours to construct 1.91 million data points. The cost of synthesizing one data point is calculated as:

$$\frac{0.35 \times 8 \times 84}{1910000} \approx 0.000123 \, \$$$

In terms of 0.01 cents, the synthesis cost is 1.23.

If we were to synthesize 2 million math problems and solutions, it would cost \$44000 using GPT-4, \$4600 using GPT-3.5, but only \$246 using GSDP. This gap becomes even more pronounced as the data volume increases.

## C  RESULTS ON SCIENTIFIC REASONING TASKS

In addition to mathematical reasoning, we utilize several widely used datasets to assess the scientific reasoning capabilities of models in subjects such as physics, biology, chemistry, and computer science. Each of these datasets is designed to challenge the models in different aspects of reasoning.

- **ARC-C** (Clark et al., 2018): ARC includes questions derived from various grade-level science exams, testing models' ability to handle both straightforward and complex scientific queries. We use the challenge subset, which contains 1,172 test questions.

- **MMLU-STEM** (Hendrycks et al., 2021): Spanning 57 subjects across multiple disciplines, MMLU evaluates the breadth and depth of a model's knowledge in a manner akin to academic and professional testing environments. We select the STEM subset of MMLU with 3.13K problems.

- **GPQA** (Rein et al., 2023): This dataset provides "Google-proof" questions in biology, physics, and chemistry, designed to test deep domain expertise and reasoning under challenging conditions. We use the diamond subset containing 198 hard problems.

- **BIG-Bench Hard (BBH)** (Suzgun et al., 2022): Consisting of 23 tasks previously found challenging for language models from BIG-Bench (Srivastava et al., 2023), BBH contains a total of 6511 challenging problems examining the capability of LLMs to solve them.

- **TheoremQA** (Chen et al., 2023b): Focused on applying mathematical theorems to solve advanced problems in fields such as mathematics, physics, and engineering, TheoremQA includes 800 questions that test the theoretical reasoning capabilities.

---

[4]https://openai.com/api/pricing/
[5]https://power.netmind.ai/rentIntro

- **MBPP** (Austin et al., 2021): MBPP consists of around 1,000 crowd-sourced Python programming problems, designed to be solvable by entry-level programmers, covering programming fundamentals, standard library functionality, and so on. Each problem consists of a task description, code solution and 3 automated test cases.

As shown in the Table 5, although our model was trained using only GSDP-MATH, it can be observed that GSDP-Qwen-7B, GSDP-7B and GSDP-8B show average improvements of 5.1%, 6.5%, and 3.5% respectively in scientific reasoning tasks. They are also highly competitive on multiple benchmarks compared to other mathematical models.

Table 5: Main results on scientific reasoning tasks. The $\Delta$ rows highlight the improvements of the GSDP models over their corresponding baseline models. The **bold** and underlined values denote the first and second best results, respectively. We employed testing scripts provided by MAmmoTH2 (Yue et al., 2024b) and OpenCompass (Contributors, 2023).

| Model | ARC-C | MMLU-STEM | GPQA | BBH | TheoremQA | MBPP | AVG |
|---|---|---|---|---|---|---|---|
| Mistral-7B | 74.2 | 50.1 | 24.7 | 55.7 | 19.2 | 47.5 | 45.2 |
| LLaMA3-8B | 78.6 | 55.6 | 27.2 | 61.1 | 20.1 | 54.9 | 49.6 |
| Qwen1.5-7B | 75.6 | 45.5 | 26.7 | 45.2 | 14.2 | 52.1 | 43.2 |
| WizardMath-7B-V1.1 | 78.3 | 54.4 | 30.8 | 57.5 | 21.1 | 56.4 | 49.8 |
| MAmmoTH-7B-Mistral | 72.1 | 48.1 | 25.3 | 48.5 | **31.5** | 46.7 | 45.4 |
| MetaMath-Mistral-7B | 76.7 | 53.0 | 28.8 | 51.2 | 19.1 | 46.3 | 45.9 |
| MathScale-Mistral | 77.3 | 54.9 | **35.4** | 56.8 | 20.8 | 54.0 | 49.9 |
| GSDP-Qwen-7B | 79.2 | 56.3 | 29.8 | 50.3 | 21.6 | 52.5 | 48.3 |
| $\Delta$ Qwen1.5-7B | +3.6 | +10.8 | +3.1 | +5.1 | +7.4 | +0.4 | +5.1 |
| GSDP-7B | 78.8 | 58.3 | 32.3 | 60.3 | 25.6 | 54.8 | 51.7 |
| $\Delta$ Mistral-7B | +4.6 | +8.2 | +7.6 | +4.6 | +6.4 | +7.3 | +6.5 |
| GSDP-8B | **80.5** | **60.8** | 30.8 | **63.7** | 24.2 | **58.4** | **53.1** |
| $\Delta$ LLaMA3-8B | +1.9 | +5.2 | +3.6 | +2.6 | +4.1 | +3.5 | +3.5 |

# D   DUAL FILTERING AND KP EXAMPLES

## D.1   DUAL FILTERING

Ensuring the quality of KPs is crucial, as using meaningless KPs can result in low-quality synthesized problems, while using overly similar KPs can lead to duplicated problems. These issues increase the computational and time costs for both problem synthesis and problem quality validation. We employ a dual filtering strategy using both embedding models and LLMs to remove low-quality and duplicated KPs. The three main steps are as follows:

- **Eliminating Low-Quality KPs:** LLMs are used to filter out KPs that are vague, contain mathematical errors, or are overly detailed. This is because vague KPs can be too broad in meaning, failing to standardize the model's output effectively. Erroneous KPs may lead the model to synthesize incorrect questions, while overly detailed KPs can overly constrain the model's output.

- **Categorization:** We first use an embedding model to calculate pairwise similarity scores between KPs. KPs with similarity scores between 0.90 and 1.0 are deemed to have the same meaning, while those with scores between 0.70 and 0.90 undergo an additional check by the LLM to confirm if they are truly synonymous. KPs with scores below 0.70 are treated as distinct. Based on this process, KPs are grouped into classes with similar KPs placed in the same class. These thresholds were determined through an analysis of the KP set.

- **Summarization:** For each KP class, the LLM identifies the most representative KP to act as the class representative. If no existing KP in the class is suitable, the LLM synthesizes a new KP to represent the class.

When only the embedding model was used for de-duplication, the quality check revealed that only 26% of the synthesized problems met the quality standard. After introducing dual filtering with the LLM, this proportion increased to 45%. This demonstrates that the dual filtering process significantly improves dataset quality while reducing problem synthesis costs.

### D.2 EXAMPLES OF BAD KNOWLEDGE POINTS

The LLM helps the embedding model classify KPs that appear similar but actually have different meanings. For example:

- *"Geometric sequence"* vs. *"Arithmetic sequence"* (similarity score: 0.805)
- *"Sine function in trigonometry"* vs. *"Cosine function in trigonometry"* (similarity score: 0.865)

The LLM effectively removes vague, mathematically incorrect, or overly detailed KPs. For example:

- Vague KPs:
  - *"Problem-solving strategies"*
  - *"Mathematical techniques"*
- Mathematically Incorrect KPs:
  - *"The sum of the outer angles of a polygon depends on the number of sides"*
  - *"The matrix result of multiplying a matrix by its inverse is the matrix itself"*
  - *"A series converges if its terms approach zero."*
  - Some incorrect or incomplete KPs
- Overly Detailed KPs:
  - *"Solving the quadratic equation $x^2 + 5x + 6 = 0$ by factoring..."*
  - Some specific problems

### D.3 EXAMPLES OF KNOWLEDGE POINTS

To demonstrate the diversity and comprehensiveness of our knowledge base, we randomly sampled 20 KPs:

*"Angle of Rotation"*, *"The unit circle and its properties"*, *"Solving Equations with Multiple Variables"*, *"Right triangles in a sphere"*, *"Inversions in permutations"*, *"Pi ($\pi$) as a constant in geometry and trigonometry"*, *"Perfect Cubes"*, *"Area of Triangles and Squares"*, *"Diophantine Approximation"*, *"Perimeter of a triangle"*, *"Abundant Number"*, *"Graphing a hyperbola"*, *"Determining the base and height of a Parallelogram"*, *"Difference of cosines formula"*, *"Quartic Polynomial"*, *"Polynomial Inequalities"*, *"Congruence of Integers"*, *"Solving equations involving digits"*, *"Sign Analysis"*, *"Calculation of expected value for a fair eight-sided die"*.

## E QUANTITATIVE ANALYSIS

We compared GSDP-MATH with MetaMath, MathCoder, and MathScale, which are open source datasets (detailed information on these datasets can be found in Table 6), in terms of seed similarity and data diversity.

### E.1 QUANTITATIVE ANALYSIS OF SEED SIMILARITY

we employed the embedding model (Chen et al., 2023a) to measure the similarity between synthetic data and seed. Specifically, for each synthetic data instance, we calculated its closest match in the seed data and recorded the highest similarity score. By aggregating the similarity scores for all synthetic data instances, we plotted histograms (Figure 5, 6, 7 and 8 ) to visualize the distribution of similarity scores and analyze the similarity between the datasets.

The figures indicate that the similarity scores for MathScale predominantly fall within the range of 0.8 to 0.9, while those for MetaMathQA and MathCodeInstruct are concentrated around 1. In

contrast, the similarity scores between GSDP-MATH and the seed data predominantly fall within the range of 0.55 to 0.65. This is because MetaMath and MathCoderInstruct rely heavily on seed data, resulting in synthesized data that is very similar to the seed data. MathScale synthesizes data based on knowledge points, which reduces overall similarity and creates a more uniform data distribution. However, because it does not fully explore implicit relationships between knowledge points (non-co-occurrence knowledge points), there is still a significant amount of data similar to the seed data. In contrast, GSDP takes into account both explicit relationships (co-occurrence knowledge points) and thoroughly explores implicit relationships. This allows our method to synthesize datasets that have a more uniform distribution and lower overall similarity, with almost no data being very similar to the seed data.

## E.2  QUANTITATIVE ANALYSIS OF DATA DIVERSITY

Given that the smallest dataset among the four contains 80K samples and the overall data volume is relatively large, we uniformly and randomly sampled 80K instances from each dataset and computed their embeddings. For each subset of embeddings, we performed clustering and compared the number of cluster centers to assess the differences in data diversity across datasets. We adopted the density-based DBSCAN (Ester et al., 1996) algorithm and utilized the k-distance graph to determine key DBSCAN parameters, ensuring a more scientifically grounded adaptation to the characteristics of each dataset.

As shown in Table 6, our method resulted in the greatest number of cluster centers, indicating higher diversity within the GSDP-MATH dataset. This finding highlights the richness of our synthetic data. Synthesizing data based on seed data or co-occurrence knowledge points often results in problems of the same type as the seed data. In contrast, our method generates new types of problems, thereby increasing the diversity of problem types in our dataset.

Table 6: Information of key attributes across various datasets. The "Sample Size" column indicates the number of instances sampled from each dataset for clustering analysis, and "Number of Clustering Centers" represents the number of distinct clusters identified in the dataset.

| Method | Seed | Synthesized Data | Size | Sample Size | Number of Clustering Centers |
|---|---|---|---|---|---|
| MetaMath | GSM8K+MATH | MetaMathQA | 395K | 80K | 339 |
| MathCoder | GSM8K+MATH | MathCodeInstruct | 80K | 80K | 271 |
| MathScale | MWPBENCH | MathScale | 2M | 80K | 488 |
| GSDP | MATH | GSDP-MATH | 1.9M | 80K | **541** |

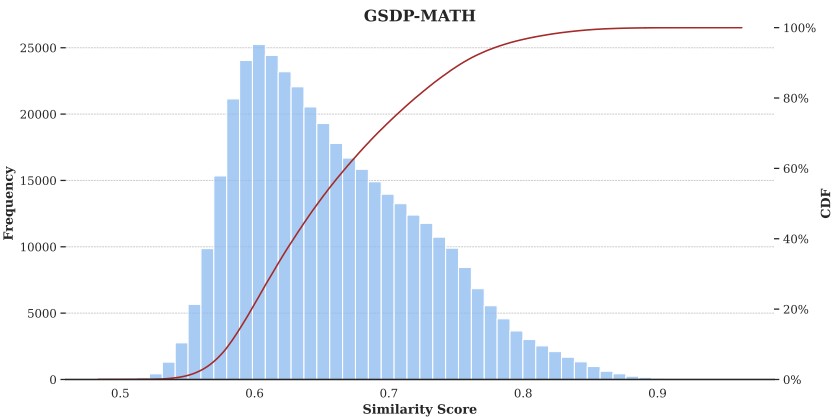

Figure 5: Histogram of the similarity scores between GSDP-MATH and the seed data. The blue bars represent the frequency of the similarity scores, while the red line represents the cumulative distribution function (CDF) of the scores.

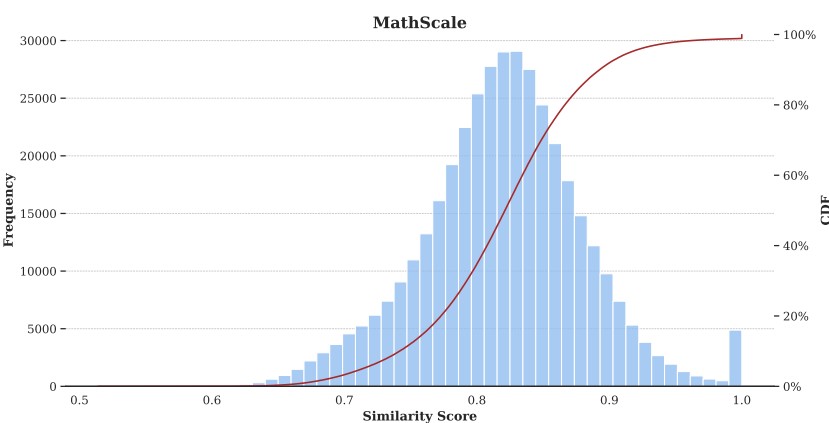

Figure 6: Histogram of the similarity scores between MathScale and the seed data. The blue bars represent the frequency of the similarity scores, while the red line represents the cumulative distribution function (CDF) of the scores.

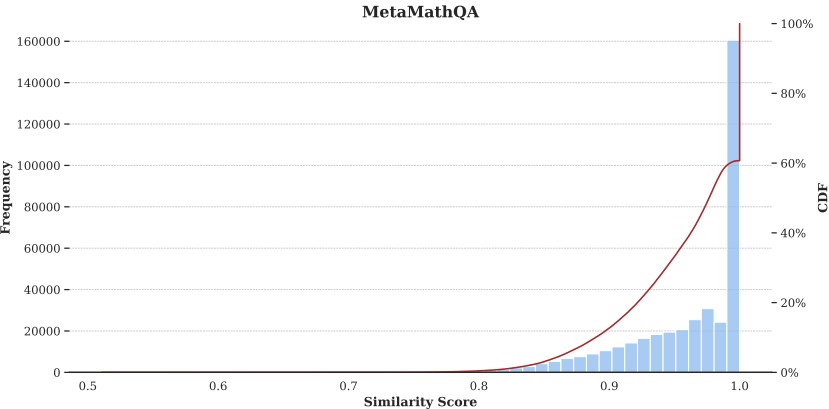

Figure 7: Histogram of the similarity scores between MetaMathQA and the seed data. The blue bars represent the frequency of the similarity scores, while the red line represents the cumulative distribution function (CDF) of the scores.

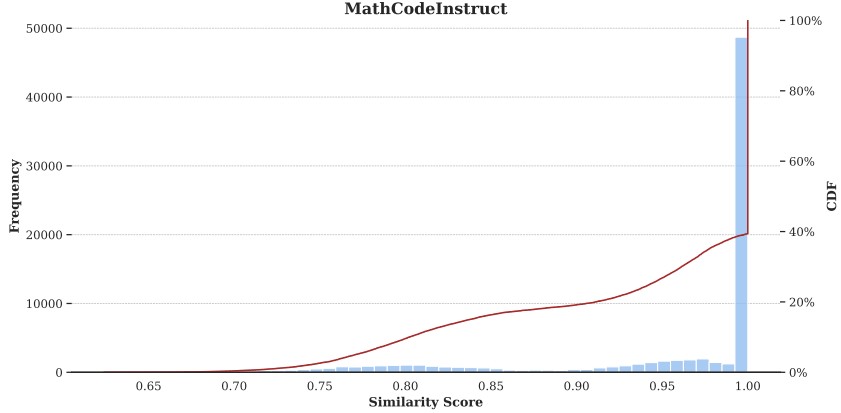

Figure 8: Histogram of the similarity scores between MathCodeInstruct and the seed data. The blue bars represent the frequency of the similarity scores, while the red line represents the cumulative distribution function (CDF) of the scores.

