# OpenReview forum: "A Graph-Based Synthetic Data Pipeline for Scaling High-Quality Data"
_ICLR.cc/2025/Conference — Submitted to ICLR 2025_

### Official Review · Reviewer_dEUx · 2024-11-04

**Soundness:** 2
**Presentation:** 3
**Contribution:** 3
**Rating:** 6
**Confidence:** 3

**Summary:**

This paper proposes the Graph-based Synthetic Data Pipeline (GSDP), a scalable and cost-effective framework for synthesizing high-quality data by leveraging knowledge graphs to expand data and reveal implicit relationships. GSDP, shown to generate synthesis quality comparable to GPT-4-0613 at a fraction of the cost, achieves strong performance in mathematical reasoning tasks, with the GSDP-7B model reaching 37.7% accuracy on MATH and 78.4% on GSM8K.

**Strengths:**

1. strong experimental performance
2. interesting research problem

**Weaknesses:**

1. The motivation is not clear enough, authors point out the limitations of existing method, including limited scalability, high cost, and similarity to seed data. However, the similarity to seed data remains questionable and lack a quantitative investigation. Moreover, why the graph-based synthetic method can solve these limitations is not very clear in the introduction section.
2. In the KP extraction process, did authors check the quality of the generated KPs and the impact of each filter and clustering operation to the quality of KPs.
3. Would be nice to conduct study to present the diversity of the generated dataset is better than other synthetic methods.
4. The baseline results based on Qwen and Llama-3 models are missing, would be nice to present these results.
5. Would be nice to conduct case study to show how the KP graph works and show the superiority compared to existing methods.

**Questions:**

please see the weaknesses above.

---

> ### Author Response · Authors · 2024-11-18
> **Question 2: In the KP extraction process, did authors check the quality of the generated KPs and the impact of each filter and clustering operation to the quality of KPs.**
>
> ## Question 2: In the KP extraction process, did authors check the quality of the generated KPs and the impact of each filter and clustering operation to the quality of KPs
>
> ---
>
> In Section 2.2 of the paper, we provided an overview of the Knowledge Points (KPs) quality filtering process. Below, we offer a detailed explanation:
>
>
>
> ### **Dual Filtering Process**
>
> Ensuring the quality of KPs is crucial, as using erroneous KPs can result in low-quality synthesized problems, while using overly similar KPs can lead to duplicated problems. These issues increase the computational and time costs for both problems synthesis and problems quality validation. As mentioned in the paper, we employ a dual filtering approach using both the embedding models and LLM to remove low-quality KPs, categorize them, and merge duplicates. The main steps are as follows:
>
> - **Eliminating Low-Quality KPs:**
>
>     LLM are used to filter out KPs that are vague, contain mathematical errors, do not adhere to proper mathematical terminology, or are overly detailed. For instance, vague KPs can be too broad in meaning, failing to standardize the model's output effectively. Erroneous KPs may lead the model to synthesize incorrect questions, while overly detailed KPs can overly constrain the model's output.
>
> - **Categorization:**
>
>     We first use an embedding model ([Bge-large-en-v1.5](https://huggingface.co/BAAI/bge-large-en-v1.5)) to calculate pairwise similarity scores between KPs. KPs with similarity scores between **0.90 and 1.0** are deemed to have the same meaning, while those with scores between **0.70 and 0.90** undergo an additional check by the LLM to confirm if they are truly synonymous. KPs with scores below **0.70** are treated as distinct. Based on this process, KPs are grouped into classes, with similar KPs placed in the same class. These thresholds were determined through an analysis of the KP set.
>
> - **Summarization:**
>
>     For each KP class, the LLM identifies the most representative KP to act as the class representative. If no existing KP in the class is suitable, the LLM synthesizes a new KP to represent the class.
>
>
>
> ### **Impact of Dual Filtering**
>
> - **Impact on Knowledge Points:**
>
>     - **De-duplication:**
>
>         The LLM helps avoid incorrect grouping of distinct KPs that the embedding model may consider overly similar. For example:
>
>          - *"Geometric sequence"* vs. *"Arithmetic sequence"* (similarity score: 0.805).
>          - *"Sine function in trigonometry"* vs. *"Cosine function in trigonometry"* (similarity score: 0.865). The LLM ensures such pairs are classified into different KP classes.
>     - **Exclusion of Low-Quality KPs:**
>
>          The LLM effectively removes vague, mathematically incorrect, or excessively complex KPs. For example:
>
>          - **Vague KPs:** Examples include *"Problem-solving strategies"* and *"Mathematical techniques"*.
>          - **Mathematically Incorrect KPs:** Examples include *"The sum of the outer angles of a polygon depends on the number of sides"*, *"The matrix result of multiplying a matrix by its inverse is the matrix itself"*, and *"A series converges if its terms approach zero."*
>          - **Overly Detailed KPs:** These typically include highly specific problem statements, such as *"Solving the quadratic equation $x^2 + 5x + 6 = 0$ by factoring……"*
>
> - **Impact on Final Results:**
>
>      - When only the embedding model was used for de-duplication, the quality check revealed that only **26%** of the synthesized problems met the quality standard. After introducing dual filtering with the LLM, this proportion increased to **45%**. This demonstrates that the dual filtering process significantly improves dataset quality while reducing problem synthesis costs.
>
>
>
> ### **Examples of Knowledge Points**
> Finally, to demonstrate the diversity and comprehensiveness of our knowledge base, we randomly sampled 20 KPs:
>
> *"Angle of Rotation"*, *"The unit circle and its properties"*, *"Solving Equations with Multiple Variables"*, *"Right triangles in a sphere"*, *"Inversions in permutations"*, *"Pi ($\pi$) as a constant in geometry and trigonometry"*, *"Perfect Cubes"*, *"Area of Triangles and Squares"*, *"Diophantine Approximation"*, *"Perimeter of a triangle"*, *"Abundant Number"*, *"Graphing a hyperbola"*, *"Determining the base and height of a Parallelogram"*, *"Difference of cosines formula"*, *"Quartic Polynomial"*, *"Polynomial Inequalities"*, *"Congruence of Integers"*, *"Solving equations involving digits"*, *"Sign Analysis"*, *"Calculation of expected value for a fair eight-sided die"*.

---

> ### Author Response · Authors · 2024-11-19
> **Question 1&3**
>
> We compared GSDP-MATH with MetaMath, MathCoder, and MathScale, which are open source datasets (detailed information on these datasets can be found in Table 1), in terms of **seed similarity** and **data diversity**, as well as the supplement to the graph-based synthetic method:
> - **Quantitative Analysis of Seed Similarity**:
>     - **Methodology**: We employed the embedding model ([Bge-large-en-v1.5](https://huggingface.co/BAAI/bge-large-en-v1.5)) to measure the similarity between synthetic data and seed data. Specifically, for each synthetic data instance, we calculated its closest match in the seed data and recorded the highest similarity score. By aggregating the similarity scores for all synthetic data instances, we plotted histograms to visualize the distribution of similarity scores and analyze the similarity between the datasets. (**We encourage the reviewer to refer to Appendix E (Figure 5, 6, 7 and 8) of the revision, where four histograms provide a comprehensive comparison of "seed similarity" across datasets.**)
>     - **Results**: The results indicate that the similarity scores between GSDP-MATH and the seed data predominantly fall within the range of 0.55 to 0.65. In contrast, the similarity scores for MathScale predominantly fall within the range of 0.8 to 0.9, while those for MetaMathQA and MathCodeInstruct are concentrated around 1. This demonstrates that our synthetic data exhibits lower similarity to the seed data, whereas the other methods generate data that is highly similar to their respective seed datasets.
>     - **Analysis**: This is because MetaMath and MathCoderInstruct rely heavily on seed data, resulting in synthesized data that is very similar to the seed data. MathScale synthesizes data based on knowledge points, which reduces overall similarity and creates a more uniform data distribution. However, because it does not fully explore implicit relationships between knowledge points (non-co-occurrence knowledge points), there is still a significant amount of data similar to the seed data. In contrast, GSDP takes into account both explicit relationships (co-occurrence knowledge points) and thoroughly explores implicit relationships. This allows our method to synthesize datasets that have a more uniform distribution and lower overall similarity, with almost no data being very similar to the seed data.
>
>
> - **Quantitative Analysis of Data Diversity**:
>     - **Methodology**: Given that the smallest dataset among the four contains 80K samples and the overall data volume is relatively large, we uniformly and randomly sampled 80K instances from each dataset and computed their embeddings. For each subset of embeddings, we performed clustering and compared the number of cluster centers to assess the differences in data diversity across datasets. We adopted the density-based DBSCAN algorithm and utilized the k-distance graph to determine key DBSCAN parameters (e.g., ε), ensuring a more scientifically grounded adaptation to the characteristics of each dataset.
>     - **Results**: As shown in Table 1, our method resulted in a greater number of cluster centers, indicating higher diversity within the GSDP-MATH dataset. This finding highlights the richness of our synthetic data.
>     - **Analysis**: Synthesizing data based on seed data or co-occurrence knowledge points often results in problems of the same type as the seed data. In contrast, our method generates new types of problems, thereby increasing the diversity of problem types in our dataset.
>
>
>     *Table 1: Information of key attributes across various datasets. The "Sample Size" column indicates the number of instances sampled from each dataset for clustering analysis, and "Number of Clustering Centers" represents the number of distinct clusters identified in the dataset.*
>
>     | Method     | Seed        | Synthesized Data | Size | Sample Size  | Number of Clustering Centers |
>     |------------|-------------|------------------|-------|-----|------------------------------|
>     | MetaMath   | GSM8K+MATH  | MetaMathQA       | 395K   | 80K      | 339                          |
>     | MathCoder  | GSM8K+MATH  | MathCodeInstruct        | 80K   | 80K      | 271                          |
>     | MathScale  | MWPBENCH    | MathScale        | 2M     | 80K      | 488                          |
>     | GSDP       | MATH        | GSDP-MATH        | 1.9M    | 80K       | 541                          |

---

> > ### Author Response · Authors · 2024-11-19
> > **Question 1&3**
> >
> > - **Supplementary Explanation on How the Graph-Based Synthesis Method Addresses These Limitations:**
> >
> >     The highlight of the graph-based synthesis method is its ability to explore both explicit and implicit relationships among knowledge points, addressing several key limitations:
> >
> >     - **High Expansion Ratio:**
> >
> >         Previous methods either relied entirely on seed data or focused solely on explicit relationships among knowledge points (co-occurrence knowledge points). However, since the seed data and the explicit relationships within it are inherently limited, these methods struggle to generate a large volume of synthesized data. In contrast, our method leverages the Knowledge Points Relationship Graph (KPRG) to uncover numerous new and reasonable knowledge point combinations by exploring implicit relationships. This allows us to achieve a significantly higher expansion ratio.
> >
> >     - **Low Seed Similarity:**
> >
> >         Methods that rely solely on seed data or explicit relationships typically generate variants of seed data problems, leading to a high degree of similarity with the seed data, as explicit relationships are already present in the seed data. Our method, however, synthesizes data by utilizing both explicit relationships and novel implicit relationships that do not appear in the seed data. This approach results in a diverse set of synthesized data with significantly lower similarity to the seed data.

---

> ### Author Response · Authors · 2024-11-19
> **Question 4**
>
> We selected Mistral-7B, a baseline model most commonly used in previous methods, for comparison with other approaches. Additionally, we chose two relatively new baseline models (LLaMA3-8B and Qwen1.5-7B) to demonstrate the versatility of our method. There are few studies using these two baseline models: LLaMA3-8B is primarily used for comparisons with this year's MammoTH2-8B, and currently, no methods use Qwen1.5-7B as a benchmark model. However, our experimental results clearly show that models based on Qwen1.5-7B and LLaMA3-8B significantly outperform the baseline.
>
> Furthermore, to avoid any potential misunderstandings, we have restructured the main table. Please refer to Table 1 in the revision.

---

> ### Author Response · Authors · 2024-11-20
> **Question 5**
>
> This is a very good suggestion. Figure 2 in the paper shows a case of synthesized data, and Figure 3 illustrates the specific process of how the KPRG works but does not provide a specific case. Therefore, we have revised Figure 3 (see revision Figure 3) in the revision to include a specific case to describe how explicit and implicit knowledge point combinations are derived.
>
> Assuming our seed data contains 4 questions, each with 2 to 3 extracted knowledge points:
>
> #### Knowledge Base of Seed Data
> ```text
> Question 1 KPs
> - (A) “Angle of Rotation”
> - (B) “The unit circle and its properties”
> - (C) “Trigonometric identities”
>
> Question 2 KPs
> - (A) “Angle of Rotation”
> - (D) “Properties of Rotational Symmetry”
>
> Question 3 KPs
> - (D) “Properties of Rotational Symmetry”
> - (E) “𝜋 as a constant in geometry and trigonometry”
>
> Question 4 KPs
> - (E) “𝜋 as a constant in geometry and trigonometry”
> - (F) “Euler's formula”
> ```
>
>
>
> We can construct the KPRG in the middle of Figure 3 from the co-occurrence of KPs. According to our definition, we then identify the core knowledge points and the combinations of knowledge points with one-hop, two-hop, three-hop, and community relationships:
>
> #### Knowledge Point Combinations
> ```text
> - Core knowledge point: A
> - One-hop: A-B, A-C, B-C, A-D, D-E, E-F
> - Two-hop: A-E, B-D, C-D, D-F
> - Three-hop: A-F
> - Community: A-B-C
> ```
>
> We can see that implicit knowledge point combinations identified through KPRG can also be used to synthesize high-quality problems that are not present in the seed data. For example:
> ```text
> A-E: The combination of "Angle of Rotation" and "Pi (π)" can be used to construct problems involving the calculation of angles in radians and the application of rotational transformations. An example problem might require students to convert angles from degrees to radians and perform rotations on a coordinate plane.
>
> C-D: The combination of "Trigonometric identities" and "Properties of Rotational Symmetry" can be used to create problems that involve proving or utilizing trigonometric identities within the context of rotational symmetry. For instance, a problem might ask students to demonstrate how certain trigonometric identities hold true under rotational transformations.
>
> A-F: The combination of "Angle of Rotation" and "Euler's formula" allows for the construction of problems that connect angular rotations to complex exponential functions. An example could involve students using Euler's formula to represent a rotation in the complex plane and interpret the geometric implications of the formula.
>
> ...
> ```
>
> We use all the identified knowledge point combinations and the following prompt as input, allowing the mathematical model to synthesize new problems. After quality checks, we obtain the GSDP-MATH.
>
> #### Prompt for Knowledge Points Extraction
> ```text
> You are a math teacher. Now, you need to help your students learn the following math knowledge points. Using these knowledge points as guidelines, please construct a new, original math problem that requires an understanding and application of all these points.
> Ensure the following:
> 1. The constructed problem must be free from any mathematical logic errors.
> 2. The problem must combine all the knowledge points.
> 3. The question should be of sufficient difficulty and logically consistent.
>
> knowledge points 1: {knowledge points1}
> knowledge points 2: {knowledge points2}
> [knowledge points 3: {knowledge points3}]
>
> Please format your response like this:
> New Problem: {Your new problem here}
> Reason: {Your explanation here}
> ```

---

> ### Author Response · Authors · 2024-11-20
> **Acknowledgement**
>
> Finally, we would like to express our gratitude for your valuable suggestions. Based on your feedback, we have made revisions in the updated version (including the introduction, Table 1, Figure 3, etc.), which are marked in blue. Additionally, we have added Appendices D and E for case studies, KP quality assessment, and quantitative experiments on the dataset.
>
> If you have any further questions or concerns, please feel free to let us know.

---

> ### Author Response · Authors · 2024-11-25
>
> Dear reviewers,
>
> We have carefully addressed your suggestions and questions, updated the paper, and submitted the revised version. The main changes are highlighted in different colors for your convenience. We kindly request you to review the revision.
>
> As the deadline is approaching, we hope you can let us know if there are any further issues so that we can address them promptly.
>
> Thank you very much!

---

> > ### Comment · Reviewer_dEUx · 2024-11-25
> > **Response**
> >
> > Thank you for your response! I appreciate the authors' efforts in addressing my concerns and will raise my score accordingly.

---

> > > ### Author Response · Authors · 2024-11-26
> > > **Thank you for raising the score and for your support.**
> > >
> > > Thank you very much for your positive feedback and for taking the time to review our paper. We greatly appreciate your valuable comments and suggestions that have helped to improve our work. We are pleased to hear that our responses have satisfactorily addressed your concerns.
> > >
> > > Thank you for raising the score and for your support.

---

### Official Review · Reviewer_867H · 2024-11-04

**Soundness:** 3
**Presentation:** 3
**Contribution:** 3
**Rating:** 3
**Confidence:** 5

**Summary:**

This paper presents a novel approach for generating synthetic data related to mathematics and mathematical reasoning using a graph-based synthetic data pipeline (GSDP). It automatically extracts knowledge points from seed data to create a knowledge point relationship graph. Utilizing the MATH training set of 7,500 problems and answers as seeds, the GSDP-MATH dataset expands to over 1.91 million pairs of math problems and answers. The authors report achieving accuracies of 37.7% on the MATH dataset and 78.4% on GSM8K when tested with several 7B LLM models. However, the dataset and models are not available for verification.

**Strengths:**

The idea of extracting knowledge points from seed data, forming a graph to explore their relationships, and then generating training data from these compressed concepts is intriguing. It aligns well with the principles of autoencoders. I believe this is probably a good angle for presenting your approach.

**Weaknesses:**

While the idea is plausible, it requires validation. Overall, I'm uncertain how this approach compares to the recent work on Critical Plan Step Learning, which appears to demonstrate greater generalization capabilities and performs better on cross-domain tasks. In principle, once you generate 1.91 million training samples, you may lose some generalization power. Additionally, the paper contains several typos.

**Questions:**

In principle, does generating 1.91 million data samples result in a loss of generalization power?

---

> ### Author Response · Authors · 2024-11-16
> **Methods Comparison and Cross-Domain Task Analysis**
>
> **Thank you for the reviewer’s comments and feedback. Below is our detailed response:**
>
> ---
>
> ### I. Regarding the comparison between the two methods suggested by the reviewer
>
> Data-driven and reinforcement learning-based methods are two key approaches to enhancing the reasoning capabilities of large language models (LLMs). Our method improves reasoning capabilities by training the model on synthesized large-scale, high-quality, and diverse mathematical data. In contrast, the Critical Plan Step Learning (CPL) method employs reinforcement learning, combining Monte Carlo Tree Search with Step-level Advantage Preference Optimization to explore and learn critical planning steps. We believe our approach offers the following distinct advantages:
>
> 1. **Lower Complexity:**
>
>     The data-driven method is relatively straightforward, requiring neither complex algorithms nor intricate training procedures. Instead, it focuses on collecting or synthesizing large amounts of high-quality data. This simplicity makes the approach easier to implement and scale, while also streamlining the learning process for the model.
>
> 2. **Scalability:**
>
>     Our approach efficiently synthesizes large-scale datasets from a small amount of seed data. As described in the paper, we achieved a 255x expansion (from 7.5k to 191M examples). In contrast, the data construction process in reinforcement learning is more intricate, making large-scale data expansion more challenging.
>
> 3. **Generalizability:**
>
>     Our method has demonstrated its generalizability across various LLMs, including Mistral-7B, LLaMA3-8B, and Qwen1.5-7B, all of which showed improved performance. On the other hand, CPL's experimental results are limited to DeepSeek-Math-Base and do not showcase its effectiveness across different models.
>
> ---
>
> ### II. Regarding the generalization capability issue raised by the reviewer
>
> We deeply understand your concern about generalization capability and have conducted detailed experimental validations in our response to address this. Specifically, we analyzed the out-of-domain reasoning capability by dividing it into two parts: **mathematical reasoning** and **cross-domain reasoning**. Here are the details:
>
> 1. **Mathematical Reasoning:**
>
>     Our synthetic data was generated based on the training set of the Math dataset. Strictly speaking, the evaluation benchmarks (e.g., GSM8K, SVAMP, GAOKAO) are out-of-domain test datasets. As shown clearly in the main results of the paper, the model's performance on these mathematical reasoning tasks improved significantly after fine-tuning with GSDP-MATH.
>
> 2. **Cross-Domain Reasoning:**
>
>     To evaluate cross-domain reasoning capability, we employed testing scripts provided by [MAmmoTH2](https://github.com/TIGER-AI-Lab/MAmmoTH2) and [OpenCompass](https://github.com/open-compass/opencompass) to test the fine-tuned GSDP model. The GSDP model is based on Mistral-7B, LLaMA3-8B, and Qwen1.5-7B, and it is fine-tuned only with GSDP-MATH. We evaluated the model across multiple benchmarks, including ARC-C, GPQA, BBH, MMLU-stem, TheoremQA, and MBPP. As shown in the table 1, the model did not experience any decline in cross-domain reasoning capability due to large-scale data training. On the contrary, it achieved performance improvements across multiple benchmarks.
>
> *Table 1: Main results on in-domain and out-of-domain reasoning tasks. The Δ rows highlight the improvements of the GSDP models over their corresponding baseline models.*
>
> | Model          | MATH | GSM8K | ARC-C | MMLU-stem | GPQA | BBH  | TheoremQA | MBPP |
> |----------------|------|-------|-------|-----------|------|------|-----------|------|
> | Mistral-7B     | 11.2 | 36.2  | 74.2  | 50.1  | 24.7 | 55.7 | 19.2   | 47.5 |
> | GSDP-7B        | 37.7 | 78.4  | 78.8  | 58.3 | 32.3 | 60.3 | 25.6   | 54.8 |
> | *Δ Mistral-7B*  | *+26.5* | *+42.2* | *+4.6* | *+8.2* | *+7.6* | *+4.6* | *+6.4* | *+7.3* |
> | LLaMA3-8B      | 21.3 | 54.8  | 78.6  | 55.6  | 27.2 | 61.1 | 20.1    | 54.9 |
> | GSDP-8B        | 37.2 | 76.5  | 80.5  | 60.8      | 30.8 | 63.7 | 24.2   | 58.4 |
> | *Δ LLaMA3-8B*  | *+15.9* | *+21.7* | *+1.9* | *+5.2* | *+3.6* | *+2.6* | *+4.1* | *+3.5* |
> | Qwen1.5-7B     | 13.3 | 54.1  | 75.6  | 45.5      | 26.7 | 45.2 | 14.2      | 52.1 |
> | GSDP-Qwen-7B   | 36.8 | 73.4  | 79.2  | 56.3      | 29.8 | 50.3 | 21.6   | 52.5 |
> | *Δ Qwen1.5-7B* | *+23.5* | *+19.3* | *+3.6* | *+10.8* | *+3.1* | *+5.1* | *+7.4* | *+0.4* |
>
> In summary, large-scale, high-quality, and diverse training data genuinely enhance the model’s reasoning capabilities across various domains. This conclusion is strongly supported by the quantitative evaluation results, further confirming that our research direction of exploring better data synthesis methods is both correct and highly meaningful.
>
> ---
> ### III. Regarding typos errors
>
> We sincerely apologize for the typos errors due to our oversight and will release a revised version to correct all inaccuracies.

---

> ### Author Response · Authors · 2024-11-20
> **Acknowledgement**
>
> Finally, we would like to express our sincere gratitude for your valuable feedback. Based on your suggestions, we have made revisions in Section 3.6 regarding the testing of model generalization ability, highlighted in brown. Additionally, we have included Appendix C to provide detailed explanations of the experiments.
>
> If you have any further questions or concerns, please feel free to let us know.

---

> ### Author Response · Authors · 2024-11-24
> **Table Update**
>
> To better showcase the advantages of the GSDP-Model, we have updated the table. Please refer to Appendix C of the revision for a more detailed table.
>
> ---
> As shown in the Table 1, although our model was trained using only GSDP-MATH, it can be observed that GSDP-Qwen-7B, GSDP-7B and GSDP-8B show average improvements of 6.5\%, 3.5\%, and 5.1\% respectively in scientific reasoning tasks. And it is highly competitive on multiple benchmarks compared to other mathematical models.
>
>
> *Table 1: Main results on scientific reasoning tasks. The Δ rows highlight the improvements of the GSDP models over their corresponding baseline models.*
>
> | Model                | ARC-C | MMLU-stem | GPQA | BBH  | TheoremQA  | MBPP         | AVG  |
> |----------------------|-------|-----------|------|------|------------|--------------|------|
> | Mistral-7B           | 74.2  | 50.1      | 24.7 | 55.7 | 19.2       | 47.5         | 45.2 |
> | LLaMA3-8B            | 78.6  | 55.6      | 27.2 | _61.1_ | 20.1     | _54.9_       | 49.6 |
> | Qwen1.5-7B           | 75.6  | 45.5      | 26.7 | 45.2 | 14.2       | 52.1         | 43.2 |
> | WizardMath-7B-V1.1   | 78.3  | 54.4      | 30.8 | 57.5 | 21.1       | 56.4         | 49.8 |
> | MAmmoTH-7B-Mistral   | 72.1  | 48.1      | 25.3 | 48.5 | **31.5**   | 46.7         | 45.4 |
> | MetaMath-Mistral-7B  | 76.7  | 53.0      | 28.8 | 51.2 | 19.1       | 46.3         | 45.9 |
> | MathScale-Mistral    | 77.3  | 54.9      | **35.4** | 56.8 | 20.8    | 54.0         | 49.9 |
> | GSDP-Qwen-7B         | _79.2_ | 56.3   | 29.8 | 50.3   | 21.6      | 52.5         | 48.3 |
> | `Δ Qwen1.5-7B`         | `+3.6`    | `+10.8`       | `+3.1`   | `+5.1`  | `+7.4`        | `+0.4`   | `+5.1`   |
> | GSDP-7B              | 78.8  | _58.3_  | _32.3_   | 60.3   | _25.6_  | 54.8         | _51.7_       |
> | `Δ Mistral-7B`         | `+4.6`    | `+8.2`        | `+7.6`   | `+4.6`  | `+6.4`        | `+7.3`   | `+6.5`   |
> | GSDP-8B              | **80.5**  | **60.8**  | 30.8 | **63.7** | 24.2 | **58.4** | **53.1** |
> | `Δ LLaMA3-8B`          | `+1.9`    | `+5.2`        | `+3.6`   | `+2.6`  | `+4.1`        | `+3.5`   | `+3.5`   |

---

> ### Author Response · Authors · 2024-11-25
>
> Dear reviewers,
>
> We have carefully addressed your suggestions and questions, updated the paper, and submitted the revised version. The main changes are highlighted in different colors for your convenience. We kindly request you to review the revision.
>
> As the deadline is approaching, we hope you can let us know if there are any further issues so that we can address them promptly.
>
> Thank you very much!

---

> ### Author Response · Authors · 2024-11-28
> **The Comparison Between Critical Plan Step Learning (CPL) Method Results and GSDP Model Results**
>
> We added CPL-final to the table to compare with our models, and highlighted it in red.
>
>
> *Table 1: Main results on scientific reasoning tasks. The Δ rows highlight the improvements of the GSDP models over their corresponding baseline models.*
>
> | Model                | ARC-C | MMLU-stem | GPQA | BBH  | TheoremQA  | MBPP         | AVG  |
> |----------------------|-------|-----------|------|------|------------|--------------|------|
> | Mistral-7B           | 74.2  | 50.1      | 24.7 | 55.7 | 19.2       | 47.5         | 45.2 |
> | LLaMA3-8B            | 78.6  | 55.6      | 27.2 | _61.1_ | 20.1     | _54.9_       | 49.6 |
> | Qwen1.5-7B           | 75.6  | 45.5      | 26.7 | 45.2 | 14.2       | 52.1         | 43.2 |
> | WizardMath-7B-V1.1   | 78.3  | 54.4      | 30.8 | 57.5 | 21.1       | 56.4         | 49.8 |
> | MAmmoTH-7B-Mistral   | 72.1  | 48.1      | 25.3 | 48.5 | **31.5**   | 46.7         | 45.4 |
> | MetaMath-Mistral-7B  | 76.7  | 53.0      | 28.8 | 51.2 | 19.1       | 46.3         | 45.9 |
> | MathScale-Mistral    | 77.3  | 54.9      | **35.4** | 56.8 | 20.8    | 54.0         | 49.9 |
> | `CPL-final`   | `56.1`  | `54.9`      | `34.3` | `60.5` | `- `  | `-` | `-`|
> | GSDP-Qwen-7B         | _79.2_ | 56.3   | 29.8 | 50.3   | 21.6      | 52.5         | 48.3 |
> | `Δ Qwen1.5-7B`         | `+3.6`    | `+10.8`       | `+3.1`   | `+5.1`  | `+7.4`        | `+0.4`   | `+5.1`   |
> | GSDP-7B              | 78.8  | _58.3_  | _32.3_   | 60.3   | _25.6_  | 54.8         | _51.7_       |
> | `Δ Mistral-7B`         | `+4.6`    | `+8.2`        | `+7.6`   | `+4.6`  | `+6.4`        | `+7.3`   | `+6.5`   |
> | GSDP-8B              | **80.5**  | **60.8**  | 30.8 | **63.7** | 24.2 | **58.4** | **53.1** |
> | `Δ LLaMA3-8B`          | `+1.9`    | `+5.2`        | `+3.6`   | `+2.6`  | `+4.1`        | `+3.5`   | `+3.5`   |

---

> ### Author Response · Authors · 2024-11-30
> **Further Clarifications and Acknowledgement**
>
> **If the above clarifications have addressed your concerns, we hope you might consider supporting our paper. We would greatly appreciate your support for our work. If you have any further questions, please do not hesitate to contact us. Thank you once again for your feedback and consideration!**

---

> ### Author Response · Authors · 2024-12-01
> **Request for Your Response**
>
> Dear reviewer,
>
> We have carefully addressed your suggestions and questions, updated the paper, and submitted the revised version. The main changes are highlighted in different colors for your convenience. We kindly request you to review the revision.
>
> As the deadline is approaching, we hope you can let us know if there are any further issues so that we can address them promptly.
>
> Thank you very much!

---

> ### Author Response · Authors · 2024-12-02
> **Key Improvements in the Rebuttal (The deadline is approaching, and we are eagerly looking forward to your response.)**
>
> **Key Improvements in the Rebuttal**
>
> **(The deadline is approaching, and we are eagerly looking forward to your response.)**
>
> Based on the valuable suggestions and questions from the three reviewers (coVk and dEUx supported our work), we have conducted additional experiments and included them in the revised Appendix and main text. Below is a summary of the key changes:
>
> **(1) Out-of-Domain Reasoning Ability Test**
>
> In **Section 3.6** and **Appendix C**, we tested the out-of-domain reasoning ability of the GSDP model. We selected multiple test sets from various fields (such as ARC-C, MMLU-STEM, GPQA, BBH, TheoremQA, and MBPP) to evaluate the model's reasoning capability in physics, chemistry, biology, and computer science (We call them scientific reasoning). Although the model was trained only on GSDP-MATH, it showed an average improvement of over **5%** (with a maximum improvement of **10.8%**) in scientific reasoning ability tests, indicating that GSDP-MATH not only significantly enhances the model's mathematical reasoning ability but also boosts its out-of-domain reasoning skills.
>
> **(2) Quantitative Analysis of the Dataset**
>
> In **Appendix E**, we conducted a quantitative analysis of the dataset's "seed similarity" and "data diversity." The results in Figure 5 and Table 6 of the revision show that GSDP-MATH has lower seed similarity and higher data diversity.
>
> **(3) Other Modifications**
>
> In the revision, we added a case study to better illustrate the GSDP workflow and its advantages over other methods. Additionally, we made it clearer in the introduction the importance of KPRG and how the Graph-based method addresses the issues of limited scalability, high cost, and similarity to seed data in existing methods. (**Appendix D.2, D.3, Figure 3**)
>
> ### **If the above clarifications have addressed your concerns, we hope you might consider supporting our paper. We would greatly appreciate your positive feedback on our work. Looking forward to your reply!**

---

### Official Review · Reviewer_coVk · 2024-11-05

**Soundness:** 2
**Presentation:** 2
**Contribution:** 2
**Rating:** 5
**Confidence:** 3

**Summary:**

Summary:

Training data comes at a premium when training generative language models to solve math (word) problems.  Semi-synthetic instance augmentation is commonly used in such cases.  This submission proposes math problem instance augmentation using "knowledge points".  I wasn't familiar with this term, but, going by the examples given, a "knowledge point" (KP) looks like a glossary entry of a mathematical term or concept like "Pythagorus theorem" or "completing the square".  These KPs can be connected by edges, based on whether/how they are related.  The proposed method samples small compact subgraphs of this KP-graph, and submits these to an LLM again to generate math problems that requires familiarity and expertise over these KPs.

**Strengths:**

Strengths:

* Identifies data scarcity problem in training LLMs to solve math (word) problems (but this is not unknown).
* Proposes selection of interconnected knowledge points as a way to synthesize high-quality, diverse math problems.
* Demonstrates that this form of data synthesis can be used to boost the performance of smaller language models.

**Weaknesses:**

Weaknesses:

* As in an increasing number of LLM-based papers, the "innovation" consists of creating a workflow and taking each task to an LLM with a suitably designed prompt.  E.g., a prompt is proposed for knowledge point (KP) extraction from a (problem, solution) pair.  There is very lightweight processing (based on cooccurrence) for graph formation among the KPs and the selection of small compact subgraphs from the KP-graph.  Then there is a prompt that sends this KP subset into an LLM again, asking it to generate a math problem involving those KPs.  Since LLMs are now basically wish-fulfillment machines, such LLM-based papers are incomparable to earlier NLP papers published at such venues.
* Perhaps as a partial consequence, the style of writing is quite foreign to students of pre-LLM ML and AI communities.  "Chain of thought" is itself a good example of how not to describe a "data structure" to a computer scientist, "knowledge point" perpetuates that style.
* We are not necessarily playing a number game here, but [https://paperswithcode.com/sota/math-word-problem-solving-on-math] lists the best MATH performance as 88.1 using a 72B LLM and 87.92 using GPT4-turbo.  In comparison the submission claims GPT-4-0613 is at 42.5 and gets it to 37.7 using a 7B LLM.  [https://paperswithcode.com/sota/arithmetic-reasoning-on-gsm8k] lists the best GSM8K performance as 96.7 using a 72B LLM and 96.4 using a 7B LLM.  In comparison the submission shows GPT-4-0613 at 92 and gets it to 76.5 using a 8B model.  I have no experience in how to give credit for lower performance using smaller models.

**Questions:**

Comments and suggestions:

The title can be greatly improved. "Data" is too generic. If you are focusing on math word problems, say so in  the title itself.

Fig 2 caption should be much longer and explain the point of the diagrams without depending on distant text.

"Knowledge point" is too non-standard and nebulous.  Please be specific and precise right the first time you use this term and define it in terms of bits and bytes.

Overall, this may be better suited to a NLP or AI conference than an ML conference.

---

> ### Author Response · Authors · 2024-11-20
> **Response to Questions and Weaknesses**
>
> 1. **Significance of the Research Direction**
>
>     We believe that our research direction is highly significant. Despite the rapid development of large language models (LLMs), there is still substantial room for improvement in complex reasoning tasks (e.g., mathematics, code, physics). An effective way to achieve this is by training models with large-scale, high-quality reasoning data. Due to the scarcity of high-quality data, synthetic data has become a popular method for constructing the necessary training datasets. However, existing methods for synthetic data generation face issues such as limited scalability, high costs, and high similarity to seed data. Our goal is to propose a scalable, cost-effective, and efficient method for synthetic data generation. Our experimental results demonstrate that our method is scalable, cost-effective, and significantly enhances model performance.
>
>     Although our research method differs from traditional natural language processing (NLP) approaches, this does not detract from its academic value and practical significance. On the contrary, LLMs have introduced more perspectives and opportunities to the NLP field, reflecting the continuous progress of scientific research and technological development.
>
>     We do not believe that a simple and effective method lacks innovation. In the current LLMs training context, high-quality data is very scarce, and being able to synthesize large-scale, high-quality data through a simple, effective method is a low-cost, high-yield endeavor.
>
>     Many advanced works have already recognized the importance of synthetic data, such as MetaMath [1] (ICLR2024), ToRA [2]  (ICLR2024), MathScale [3] (ICML2024), MammoTH2 [4] (NeurIPS2024), and MathCoder [5] (ICLR2024). Therefore, we believe that publishing our paper at this conference is appropriate.
>
> 2. **Regarding the term “Knowledge Point”**
>
>     The reason we use the term “Knowledge Point” in our paper is to align with previous methods that also use this term. However, your suggestion is very appreciated. We have added an explanation where “Knowledge Point” first appears in the revision and have improved the caption for Figure 2, both marked in red.
>
> 3. **Concerning the term “data” in the Title**
>
>     - **3.1. Explanation of the Term "Data"**
>       The term “data” in the title is intended to indicate multiple types of data. Our method is not limited to data from a single domain but can be applied to data from various fields, such as physics, chemistry, biology, and code. We have also stated in the introduction that complex reasoning encompasses multiple aspects, and we chose the most challenging mathematical reasoning tasks for our experiments.
>
>     - **3.2. Title Modification**
>       Your suggestion is very valuable. To avoid ambiguity and to accurately convey our intent, we have decided to change the term “data” in the title to “reasoning instructions”, while also making minor adjustments to the abstract.
>
> 4. **Regarding the Comparison of Model Performance**
>
>     As for the issue of existing high-performance models, our experiment aims to prove the effectiveness of the synthetic data generated by our method, i.e., significantly enhancing the reasoning capabilities of LLMs. We only used GSDP-MATH for training, and the comparison models were also trained using synthetic data. Therefore, reviewers need to understand that our goal is to prove the effectiveness of our synthetic method, which has significant advantages over other synthetic methods.
>
>     The high-performance models you mentioned were trained with a vast amount of rich mathematical data; thus, our models cannot be directly compared to these commercial-grade mathematical models. The key contribution of our paper lies in proposing an efficient data synthesis method for the industry, hoping to provide data support for other researchers in their LLM training endeavors.
>
> ---
>
> [1] Longhui Yu, Weisen Jiang, Han Shi, Jincheng Yu, Zhengying Liu, Yu Zhang, James T Kwok, Zhenguo Li, Adrian Weller, and Weiyang Liu. Metamath: Bootstrap your own mathematical questions for large language models. arXiv preprint arXiv:2309.12284, 2023.
>
> [2] Gou, Z., Shao, Z., Gong, Y., Yang, Y., Huang, M., Duan, N., Chen, W., et al. Tora: A tool-integrated reasoning agent for mathematical problem solving. arXiv preprint arXiv:2309.17452, 2023.
>
> [3] Zhengyang Tang, Xingxing Zhang, Benyou Wan, and Furu Wei. Mathscale: Scaling instruction tuning for mathematical reasoning. arXiv preprint arXiv:2403.02884, 2024.
>
> [4] Xiang Yue, Tuney Zheng, Ge Zhang, and Wenhu Chen. Mammoth2: Scaling instructions from the web. arXiv preprint arXiv:2405.03548, 2024b.
>
> [5] Ke Wang, Houxing Ren, Aojun Zhou, Zimu Lu, Sichun Luo, Weikang Shi, Renrui Zhang, Linqi
> Song, Mingjie Zhan, and Hongsheng Li. Mathcoder: Seamless code integration in llms for enhanced mathematical reasoning. arXiv preprint arXiv:2310.03731, 2023.

---

> > ### Comment · Reviewer_coVk · 2024-11-27
> > **Thanks for the clarifications**
> >
> > Generating synthetic data is of course highly motivated for many kinds of tasks.
> > There are some steps in your proposal, like screening KPs for quality, that still appear largely or fully manual.
> > Have you accounted for the cognitive workload of KP screening compared to conventional ways of generating data, or ways of helping programs synthesize data for augmentation?
> > Overall, based on your inputs, I can increase my score to weak reject, but I cannot champion the paper.
> > It seems a bit preliminary, with various loose ends still left to tie up.
> > But I appreciate your  effort with this paper!

---

> ### Author Response · Authors · 2024-11-20
> **Acknowledgement**
>
> Finally, we would like to express our gratitude for your valuable suggestions. Based on your feedback, we have made revisions in the updated version (including the introduction and Figure 1), which are marked in red. Additionally, we have added supplementary experiments in the appendix, such as the testing of model generalization ability, case studies, KP quality assessment, and quantitative experiments on the dataset.
>
> If you have any further questions or concerns, please feel free to let us know.

---

> ### Author Response · Authors · 2024-11-25
>
> Dear reviewers,
>
> We have carefully addressed your suggestions and questions, updated the paper, and submitted the revised version. The main changes are highlighted in different colors for your convenience. We kindly request you to review the revision.
>
> As the deadline is approaching, we hope you can let us know if there are any further issues so that we can address them promptly.
>
> Thank you very much!

---

> ### Author Response · Authors · 2024-11-28
> **Acknowledgements and Clarifications**
>
> Thank you very much for your positive feedback on our work. We have addressed your concerns in our responses to other reviewers. Here, I will provide a detailed reply focusing on two aspects to address some potential concerns you might have: "Automation Level of the Pipeline" and "Additional Clarifications of Our Work."
>
> ### **1. Automation Level of the Pipeline**
>
> Our entire data synthesis pipeline is fully automated, requiring only the pre-design of prompts and synthesis algorithms.
>
> **(1) Screening of KPs**
>
> Regarding your concerns about the knowledge point screening process, we have elaborated on the "dual filtering" screening strategy in Section 2.2 of our paper. After designing the prompts, input-output, and algorithm process, the entire procedure is handled by an embedding model and an LLM. Given the limited number of KPs (<10k), the process does not require significant computational time or manual intervention.
>
> Here is a brief introduction to the "dual filtering" strategy. For more details, please refer to **Section 2.2** of the revision and **Appendix D.1**.
>
> The "dual filtering" strategy leverages embedding models and LLMs to remove low-quality and duplicate knowledge points:
> - **Removal of low-quality knowledge points**: The LLM is used to filter out KPs that are ambiguous, contain mathematical errors, do not conform to mathematical standards, or are overly detailed (involving specific problem requirements).
> - **Deduplication of knowledge points**: First, the embedding model processes the KPs to calculate the similarity between them. KPs with a similarity score between 0.9 and 1.0 are considered similar; those with a similarity score between 0.7 and 0.9 need further confirmation through the LLM to exclude those that appear similar but are actually different, e.g., "Geometric sequence" (similarity score: 0.805) vs. "Arithmetic sequence," and "Sine function in trigonometry" (similarity score: 0.865) vs. "Cosine function in trigonometry." KPs with a similarity score below 0.7 are considered dissimilar. Then, we group all similar KPs into categories and use the LLM to select or generate the best representative KP for each category.
>
> **(2) Automation Level of Other Parts of the Pipeline**
>
> The extraction of knowledge points, selection of reasonable KP combinations, synthesis of problems and answers, and final quality checks in the pipeline are all completed using pre-designed prompts and synthesis algorithms, with no need for manual selection or verification. The manual verification of the reasonableness and effectiveness of prompts and algorithm designs is an essential part of every work and cannot be avoided.
>
> ### **2. Additional Clarifications of Our Work**
>
> Based on the suggestions from the other two reviewers (**one of whom, dEUx, has positively evaluated our work**), we have conducted additional experiments included in the revision's Appendix and main text. Here is a summary addressing some potential concerns:
>
> **(1) Out-of-Domain Reasoning Ability Test**
>
> In **Section 3.6** and **Appendix C**, we tested the out-of-domain reasoning ability of the GSDP model. We selected multiple test sets from various fields (such as ARC-C, MMLU-STEM, GPQA, BBH, TheoremQA, and MBPP) to evaluate the model's reasoning capability in physics, chemistry, biology, and computer science (We call them scientific reasoning). Although the model was trained only on GSDP-MATH, it showed an average improvement of over **5%** (with a maximum improvement of **10.8%**) in scientific reasoning ability tests, indicating that GSDP-MATH not only significantly enhances the model's mathematical reasoning ability but also boosts its out-of-domain reasoning skills.
>
> **(2) Quantitative Analysis of the Dataset**
>
> In **Appendix E**, we conducted a quantitative analysis of the dataset's "seed similarity" and "data diversity." The results in Figure 5 and Table 6 of the revision show that GSDP-MATH has lower seed similarity and higher data diversity.
>
> **(3) Other Modifications**
>
> In the revision, we added a case study to better illustrate the GSDP workflow and its advantages over other methods. Additionally, we made it clearer in the introduction the importance of KPRG and how the Graph-based method addresses the issues of limited scalability, high cost, and similarity to seed data in existing methods. (**Appendix D.2, D.3, Figure 3**)
>
> ### **If the above clarifications have addressed your concerns, we hope you might consider championing our paper. We would greatly appreciate your support for our work. Thank you once again for your feedback and consideration!**

---

> > ### Author Response · Authors · 2024-11-28
> > **Out-of-Domain Reasoning Ability Test**
> >
> > As shown in the Table 1, although our model was trained using only GSDP-MATH, it can be observed that GSDP-Qwen-7B, GSDP-7B and GSDP-8B show average improvements of 6.5\%, 3.5\%, and 5.1\% respectively in scientific reasoning tasks. And it is highly competitive on multiple benchmarks compared to other mathematical models. Please refer to Appendix C of the revision for a more detailed table.
> >
> >
> > *Table 1: Main results on scientific reasoning tasks. The Δ rows highlight the improvements of the GSDP models over their corresponding baseline models.*
> >
> > | Model                | ARC-C | MMLU-stem | GPQA | BBH  | TheoremQA  | MBPP         | AVG  |
> > |----------------------|-------|-----------|------|------|------------|--------------|------|
> > | Mistral-7B           | 74.2  | 50.1      | 24.7 | 55.7 | 19.2       | 47.5         | 45.2 |
> > | LLaMA3-8B            | 78.6  | 55.6      | 27.2 | _61.1_ | 20.1     | _54.9_       | 49.6 |
> > | Qwen1.5-7B           | 75.6  | 45.5      | 26.7 | 45.2 | 14.2       | 52.1         | 43.2 |
> > | WizardMath-7B-V1.1   | 78.3  | 54.4      | 30.8 | 57.5 | 21.1       | 56.4         | 49.8 |
> > | MAmmoTH-7B-Mistral   | 72.1  | 48.1      | 25.3 | 48.5 | **31.5**   | 46.7         | 45.4 |
> > | MetaMath-Mistral-7B  | 76.7  | 53.0      | 28.8 | 51.2 | 19.1       | 46.3         | 45.9 |
> > | MathScale-Mistral    | 77.3  | 54.9      | **35.4** | 56.8 | 20.8    | 54.0         | 49.9 |
> > | GSDP-Qwen-7B         | _79.2_ | 56.3   | 29.8 | 50.3   | 21.6      | 52.5         | 48.3 |
> > | `Δ Qwen1.5-7B`         | `+3.6`    | `+10.8`       | `+3.1`   | `+5.1`  | `+7.4`        | `+0.4`   | `+5.1`   |
> > | GSDP-7B              | 78.8  | _58.3_  | _32.3_   | 60.3   | _25.6_  | 54.8         | _51.7_       |
> > | `Δ Mistral-7B`         | `+4.6`    | `+8.2`        | `+7.6`   | `+4.6`  | `+6.4`        | `+7.3`   | `+6.5`   |
> > | GSDP-8B              | **80.5**  | **60.8**  | 30.8 | **63.7** | 24.2 | **58.4** | **53.1** |
> > | `Δ LLaMA3-8B`          | `+1.9`    | `+5.2`        | `+3.6`   | `+2.6`  | `+4.1`        | `+3.5`   | `+3.5`   |

---

> > > ### Comment · Reviewer_coVk · 2024-12-02
> > > **Many thanks for actively engaging in the rebuttal process**
> > >
> > > Your clarifications and elaboration of the workflow will make the writeup much better, and the results potentially reproducible by readers. Based on my understanding of the creativity of the harnessing LLMs to such data augmentation, I would like to hold the overall rating at the same level. All the best!

---

### Author Response · Authors · 2024-11-25

Dear reviewers,

We have carefully addressed your suggestions and questions, updated the paper, and submitted the revised version. The main changes are highlighted in different colors for your convenience. We kindly request you to review the revision.

As the deadline is approaching, we hope you can let us know if there are any further issues so that we can address them promptly.

Thank you very much!

---

### Author Response · Authors · 2024-12-02
**Key Improvements in the Rebuttal**

Based on the valuable suggestions and questions from the three reviewers (**we sincerely appreciate the support of coVk and dEUx for our work**), we have conducted additional experiments and included them in the revised Appendix and main text. Below is a summary of the key changes:

**(1) Out-of-Domain Reasoning Ability Test**

In **Section 3.6** and **Appendix C**, we tested the out-of-domain reasoning ability of the GSDP model. We selected multiple test sets from various fields (such as ARC-C, MMLU-STEM, GPQA, BBH, TheoremQA, and MBPP) to evaluate the model's reasoning capability in physics, chemistry, biology, and computer science (We call them scientific reasoning). Although the model was trained only on GSDP-MATH, it showed an average improvement of over **5%** (with a maximum improvement of **10.8%**) in scientific reasoning ability tests, indicating that GSDP-MATH not only significantly enhances the model's mathematical reasoning ability but also boosts its out-of-domain reasoning skills. **Table 5 in the revision** and **Table 1 below this comment** show the improvement of our model on scientific reasoning tasks and its comparison with other models (**including the CPL method requested by Reviewer 867H. Additionally, as the deadline is approaching, we hope Reviewer 867H can provide feedback so that we can respond as soon as possible**).

---
*Table 1: Main results on scientific reasoning tasks. The Δ rows highlight the improvements of the GSDP models over their corresponding baseline models.*

| Model                | ARC-C | MMLU-stem | GPQA | BBH  | TheoremQA  | MBPP         | AVG  |
|----------------------|-------|-----------|------|------|------------|--------------|------|
| Mistral-7B           | 74.2  | 50.1      | 24.7 | 55.7 | 19.2       | 47.5         | 45.2 |
| LLaMA3-8B            | 78.6  | 55.6      | 27.2 | _61.1_ | 20.1     | _54.9_       | 49.6 |
| Qwen1.5-7B           | 75.6  | 45.5      | 26.7 | 45.2 | 14.2       | 52.1         | 43.2 |
| WizardMath-7B-V1.1   | 78.3  | 54.4      | 30.8 | 57.5 | 21.1       | 56.4         | 49.8 |
| MAmmoTH-7B-Mistral   | 72.1  | 48.1      | 25.3 | 48.5 | **31.5**   | 46.7         | 45.4 |
| MetaMath-Mistral-7B  | 76.7  | 53.0      | 28.8 | 51.2 | 19.1       | 46.3         | 45.9 |
| MathScale-Mistral    | 77.3  | 54.9      | **35.4** | 56.8 | 20.8    | 54.0         | 49.9 |
| CPL-final   | 56.1  | 54.9      | 34.3 | 60.5 | -   | - | -|
| GSDP-Qwen-7B         | _79.2_ | 56.3   | 29.8 | 50.3   | 21.6      | 52.5         | 48.3 |
| `Δ Qwen1.5-7B`         | `+3.6`    | `+10.8`       | `+3.1`   | `+5.1`  | `+7.4`        | `+0.4`   | `+5.1`   |
| GSDP-7B              | 78.8  | _58.3_  | _32.3_   | 60.3   | _25.6_  | 54.8         | _51.7_       |
| `Δ Mistral-7B`         | `+4.6`    | `+8.2`        | `+7.6`   | `+4.6`  | `+6.4`        | `+7.3`   | `+6.5`   |
| GSDP-8B              | **80.5**  | **60.8**  | 30.8 | **63.7** | 24.2 | **58.4** | **53.1** |
| `Δ LLaMA3-8B`          | `+1.9`    | `+5.2`        | `+3.6`   | `+2.6`  | `+4.1`        | `+3.5`   | `+3.5`   |
---

**(2) Quantitative Analysis of the Dataset**

In **Appendix E**, we conducted a quantitative analysis of the dataset's "seed similarity" and "data diversity." The results in **Figure 5** and **Table 6** of the **revision** show that GSDP-MATH has lower seed similarity and higher data diversity.

**(3) Other Modifications**

In the revision, we added a case study to better illustrate the GSDP workflow and its advantages over other methods. Additionally, we made it clearer in the introduction the importance of KPRG and how the Graph-based method addresses the issues of limited scalability, high cost, and similarity to seed data in existing methods. (**Appendix D.2, D.3, Figure 3 in the revision**)

---

### Meta-Review · Area_Chair_1dBV · 2024-12-13

**Metareview:**

This paper proposes a Graph-based Synthetic Data Pipeline (GSDP) for generating synthetic data to improve mathematical reasoning capabilities in language models. The framework extracts knowledge points (KPs) from seed mathematical problems and creates a knowledge graph representing relationships between these concepts. Using the MATH training set (7,500 problems) as seeds, GSDP generated a dataset of 1.91 million problem-answer pairs (GSDP-MATH). When tested with 7B parameter LLM models, the approach achieved 37.7% accuracy on the MATH dataset and 78.4% accuracy on GSM8K. The paper claims to offer a more scalable and cost-effective approach compared to existing methods, producing synthesis quality comparable to GPT-4-0613.

The paper addresses an important challenge in the field - the scarcity of training data for mathematical reasoning tasks. The proposed approach of using interconnected knowledge points for problem synthesis is novel and theoretically well-motivated, drawing parallels to autoencoder principles. The method demonstrates the ability to boost the performance of smaller language models, potentially making mathematical reasoning more accessible with less computational resources. The experimental results show promising performance improvements, particularly considering the model size constraints.

However, the approach heavily relies on LLM-based workflows with prompt engineering, rather than introducing fundamental algorithmic innovations. The quality and impact of the KP extraction process and filtering operations are not thoroughly validated, and there is a lack of quantitative investigation into dataset diversity and comparison with other synthetic methods. The dataset and models are not available for verification, missing baseline results for important models (Qwen and Llama-3), and performance still lags behind state-of-the-art results on benchmark datasets. Additionally, there is insufficient motivation for why a graph-based approach addresses the stated limitations, lack of detailed case studies demonstrating the KP graph's effectiveness, use of non-standard terminology ("knowledge point") without precise definition, and several typos noted in the manuscript.

Based on the reviews and analysis, rejection is recommended for this paper. While it presents an interesting approach, the heavy reliance on LLM-based workflows with minimal algorithmic innovation raises concerns about the fundamental contribution to the field. The lack of rigorous validation of key components (KP extraction, filtering, dataset diversity) and missing baseline comparisons makes it difficult to fully assess the method's effectiveness. Although the performance shows promise for smaller models, it does not advance the state-of-the-art, and the inability to verify results due to unavailable data and models is concerning. The presentation and motivation issues suggest the work would benefit from substantial revision and additional experimental validation before being ready for publication. While the paper shows promise and addresses an important problem, it would benefit from addressing these limitations in a revised version, including more thorough empirical validation and clearer positioning of the technical contribution beyond LLM-based workflows.

**Additional Comments On Reviewer Discussion:**

During the rebuttal period, reviewers raised several concerns about the paper, which the authors attempted to address through their responses. R1 expressed concerns about the paper primarily presenting an LLM-based workflow rather than fundamental innovation, though the authors explained their manual screening process for KPs. While R1 acknowledged some improvements, they maintained that the work appeared preliminary.

R2 questioned the effectiveness of the approach compared to Critical Plan Step Learning (CPL). The authors responded by highlighting three main advantages of their method: lower implementation and training complexity, better scalability with 255x data expansion, and broader generalizability across various LLM architectures.

R3 initially had concerns about motivation clarity, KP extraction quality control, dataset diversity, and missing baseline results. The authors' responses satisfied R3, leading to an improved rating. Following the rebuttal, R1 maintained a "weak reject" position while acknowledging the value of the clarifications. R2 did not explicitly respond to the authors' rebuttal, while R3 increased their score based on the responses.

The authors successfully provided clear explanations of their method's advantages and improved technical reproducibility while addressing concerns about model generalizability. However, significant concerns remained, including limited methodological innovation, the necessity of manual intervention in the workflow, and the preliminary nature of the work as noted by R1. The lack of response from R2 made it difficult to assess if their concerns were adequately addressed.

Although the rebuttal period provided valuable clarifications, it did not fully address core concerns about the paper's fundamental contribution and methodological innovation. Despite some reviewers becoming more positive after the rebuttal, the remaining concerns about the work's preliminary nature and reliance on LLM-based workflows supported maintaining the original rejection recommendation.

---

### Decision · Program_Chairs · 2025-01-22

Reject